# Distributed Multi-Agent Lifelong Learning

**Prithviraj P. Tarale**                                                              *ptarale@umass.edu*
*Department of Computer Science*
*University of Massachusetts Amherst Amherst, MA 01003*

**Edward Rietman**                                                                   *erietman@umass.edu*
*Department of Computer Science*
*University of Massachusetts Amherst Amherst, MA 01003*

**Hava T. Siegelmann**                                                               *hava@umass.edu*
*Department of Computer Science*
*University of Massachusetts Amherst Amherst, MA 01003*

**Reviewed on OpenReview:** *https://openreview.net/forum?id=IIVr4Hu3Oi*

## Abstract

Lifelong learning (LL) machines are designed to operate safely in dynamic environments by continually updating their capabilities. Conventional LL paradigms often assume that new data come labeled and that each LL machine has to learn independently from its environment. However, human labeling is expensive and impractical in remote conditions where automation is most desired. We introduce the Peer Parallel Lifelong Learning (PEEPLL) framework for distributed Multi-Agent Lifelong Learning, where agents continually learn online by actively requesting assistance from other agents instead of relying on the expensive environment to teach them. Unlike classical distributed AI, where communication scales poorly, lifelong learners need to communicate only on information they have not yet learned. Additionally, agents reply only if they are highly confident: Our TRUE confidence score represents an application of a Variational Classifier to quantify confidence in prediction. TRUE outperforms traditional Entropy-based confidence scores, reducing communication overhead by 18.05% on CIFAR-100 and 5.8% on MiniImageNet. To improve system resilience to low-quality or adversarial responses, our agents selectively accept a subset of received responses using the REFINE algorithm, which results in a 51.99% increase in the percentage of correct accepted responses on CIFAR-100 and 25.79% on MiniImageNet. Like traditional LL agents, PEEPLL agents store a subset of previously acquired knowledge as memory to learn alongside new information to prevent forgetting. We propose a Dynamic Memory-Update mechanism for PEEPLL agents that improves QA's classification performance by 44.17% on CIFAR-100 and 26.8% on MiniImageNet compared to the baseline Memory-Update mechanism. Additionally, we demonstrate that a PEEPLL agent can outperform an LL agent even if the latter has environmental supervision available, thus significantly reducing the need for labeling. PEEPLL provides a framework to facilitate research in distributed multi-agent LL, marking a substantial step towards practical, scalable lifelong learning technologies at the edge. Our code is available at https://github.com/Prithvitarale/peepll.

## 1 Introduction

Lifelong Learning (LL) is an emerging field of AI that seeks to continually improve system capabilities with runtime experience while preventing catastrophic forgetting of previously learned experiences (Aljundi et al., 2019b;a; Mai et al., 2020; Chaudhry et al., 2018b; 2019). It makes no assumptions about how the data appear in the real environment. To be context-aware, LL must include online lifelong learning (OLL), where models learn online from each data point only once. This is in addition to an offline global consolidation when

time permits. OLL is particularly relevant for edge conditions where agents must be highly autonomous and adaptive.

State-of-the-art LL paradigms often assume that all incoming new data are labeled through environmental supervision and that LL agents must learn individually from their environments. This assumption is limiting the applicability of AI. In realistic environments, agents may have varied skills that they learned through their particular experiences. We propose a paradigm in which agents can learn from their peer agents in addition to learning from the environment. This improves the system's adaptability and reduces reliance on expensive environmental supervision. This study focuses on presenting and solving distributed *multi-agent* OLL, which has numerous applications, from self-driving vehicles to drones and more. Unlike classical distributed AI, our agents continually learn and communicate only during unprecedented events, progressively reducing inter-agent communication as such events become precedented; this limits communication among agents. Reducing reliance on environmental supervision is important to make LL applications more seamless for realistic and edge environments.

Recent literature motivates the urgency to incorporate a multi-agent perspective into LL paradigms. The authors who participated in DARPA's Shared Experience Lifelong Learning (ShELL) program argue that "One vision of a future artificial intelligence (AI) is where many separate units can learn independently over a lifetime and share their knowledge with each other. ... **The result is a network of agents that can quickly respond to and learn new tasks, that collectively hold more knowledge than a single agent and that can extend current knowledge in more diverse ways than a single agent.** ... We propose that the convergence of such scientific and technological advances will lead to the emergence of new types of scalable, resilient and sustainable AI systems." (Soltoggio et al., 2024). They suggest, among possible applications, Multi-Agent Active Sensing, Space Exploration, Responsive and Personalized Medicine, and Distributed Cybersecurity Systems.

Multi-Agent Machine Learning (MAML) facilitates agent collaboration either by solving problems jointly during deployment (Foerster et al., 2016; Gupta et al., 2017; Hüttenrauch et al., 2017) or by distributing tasks among agents during training and integrating their efforts during testing (Raja et al., 2022; Kim et al., 2022; Brito et al., 2021). However, the integration of MAML into LL introduces new challenges and necessitates a reevaluation of the traditional MAML and LL methods. A primary challenge in MAML settings is that the communication overhead between agents scales poorly as the number of agents increases. Furthermore, a unique limitation arises when integrating MAML into LL: communication protocols in MAML are often based on past interactions, reputations, or a fixed, known task distribution among agents (Das et al., 2018; Jiang & Lu, 2018; Hoshen, 2017; Das et al., 2017). This limiting assumption does not hold for LL agents who continuously update their knowledge. Finally, agents must navigate learning from potentially incorrect peer responses. To address these issues, we introduce a framework to facilitate multi-agent coordination for LL agents: **Pee**r, **P**arallel **L**ifelong **L**earning (PEEPLL, pronounced 'People').

## 1.1 Peer, Parallel Lifelong Learning (PEEPLL)

PEEPLL's mechanism and framework are designed to navigate the dynamic, uncertain environments typical of multi-agent LL settings; it focuses on judicious peer communication without overwhelming the system with excessive communication. Multi-agent cooperation across domains of fields helps reduce reliance on annotated data (environmental supervision). Regardless of the domain, in any decentralized multi-agent LL system, agents must be self-aware - able to autonomously identify novel tasks, actively seek help, responsibly assist others, and carefully evaluate received help.

While this study focuses on supervised image classification, Lifelong Learning is also increasingly being applied in domains like language modeling (Gururangan et al., 2020; Qin et al., 2023), robotics (Lesort et al., 2019; Logacjov Aleksej, 2021), and medical AI (Lee & Lee, 2020; Vokinger et al., 2021), where agents develop sub-domain-expertise in their respective environments through continual learning. The principles underlying PEEPLL are applicable across domains. In language modeling, for instance, PEEPLL can allow LLM agents to communicate with peers when handling uncertain prompts, allowing them to continuously learn to generate accurate and contextually relevant responses. The received peer responses can also be used for future fine-tuning. In high-stakes medical AI, PEEPLL can allow agents to seek assistance when

uncertain and take only confident and well-validated actions. In robotics, PEEPLL can facilitate real-time collaboration, enabling robots to responsibly exchange sub-domain-specific expertise and adapt to new tasks (either online or offline, depending on the application). The **Framework** section below provides domain-agnostic guidelines for developing dedicated algorithms to the components of PEEPLL.

Our solution to PEEPLL demonstrates that a PEEPLL agent with no environmental supervision can perform better on novel tasks than an LL agent with complete environmental supervision. This advancement helps minimize disruptions to the user experience and progresses toward realizing seamless ShELL technologies.

***Mechanism:*** When an agent, designated Query Agent (QA), receives a query that it cannot confidently answer, it seeks assistance from other peer agents in the network, designated Response Agents (RAs). The RAs respond if they are confident in their answers. The QA then selects a subset of these responses for lifelong learning. Please refer to Figure 1 that illustrates the PEEPLL mechanism in detail.

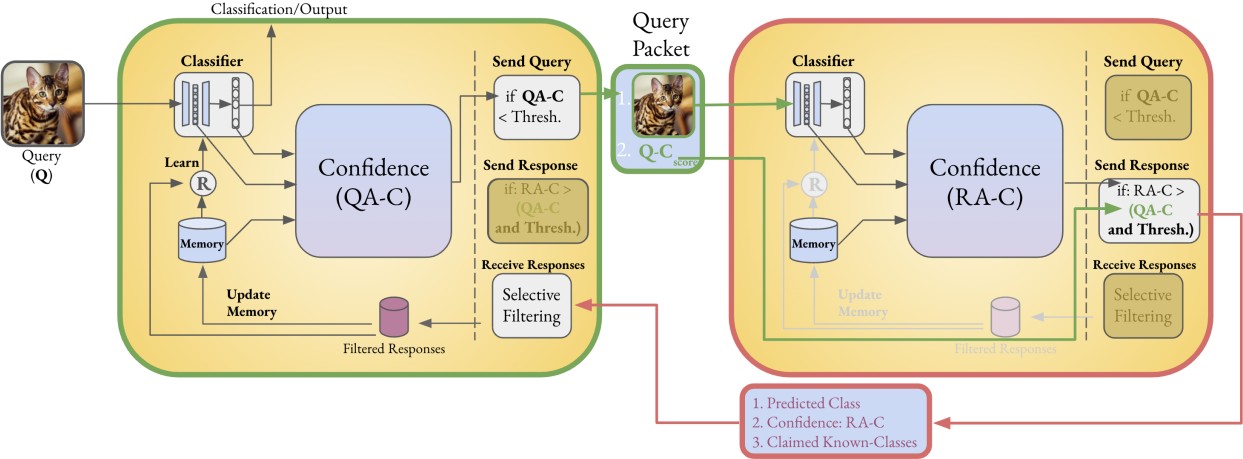

Figure 1: **PEEPLL's Query-Reponse Mechanism:** The agent receiving the original query is designated as the Query Agent (QA), while the agents it seeks assistance from are termed Response Agents (RAs). All agents are identical and can function interchangeably as either QA or RA, depending on the recipient of the query. Upon receiving a query, the QA assesses its confidence in providing an accurate response. If underconfident, the QA queries other agents in the network (RAs). RAs then evaluate their confidence in responding to the query. If confident, they transmit their responses back to the QA. Finally, the QA applies Selective Filtering to accept only a subset of the received responses. The QA then learns these accepted responses in conjunction with replay samples from its memory (circle with R) while updating its memory with the accepted responses.

***Framework:*** We identify three major components of the PEEPLL mechanism that are essential for effective multi-agent coordination in lifelong learning technologies - (1) Confidence-Evaluation, (2) Selective Filtering, and (3) Lifelong Learning in PEEPLL. In the remainder of this section, we detail the components and initial benchmarks of those components.

1. **Confidence-Evaluation:** This component enables (a) the QA to autonomously recognize novel tasks and request assistance when needed and (b) the RAs to evaluate their ability to assist the QA reliably, transmitting only likely correct and relevant responses. The primary objective is to (1) restrict communication to essential queries and responses and (2) promote stable learning at the QA.

   Given the continual evolution of LL agents' knowledge, Confidence Evaluation must operate beyond reliance on historical performance or preexisting knowledge bases, ensuring assessments are agent-agnostic and contextually relevant to each specific query. Moreover, the confidence score must retain its efficacy even after the agent has lifelong learned.

   Our proposed TRUE confidence score reduces communication overhead by 18.05% on CIFAR-100 and 5.8% on Mini-ImageNet compared to the widely accepted and effective 'Entropy' based methods

(Yona et al., 2022; Chen et al., 2021) (see Section 4.1). TRUE serves as a general contribution to the ML community for uncertainty quantification.

2. **Selective Response Filter:** After receiving responses from all the confident RAs, the QA further accepts only the most reliable responses. This component is important to security in multi-agent systems, assuring that an agent does not simply choose the first or the most confident response, which might have been adversarial.

   Our most effective method increases the proportion of correct responses in the selected responses by 51.99% on CIFAR100 and 25.79% on MiniImageNet (Section 4.2).

3. **Lifelong Learning in PEEPLL:** Integrating MAML into LL requires reevaluating traditional LL concepts. Replay-based LL agents allocate memory to store samples from previous experiences and learn alongside new information to prevent forgetting. In PEEPLL, incorporating incorrect responses from RAs into the QA's learning process and memory introduces new challenges:

   (a) *Memory-Update:* Algorithms must handle incorrect responses from RAs when updating QA's memory. These incorrect responses can fill the QA's memory budget and disturb learning, but we show that they may also have a beneficial regularization effect on learning (see Section 4.3).

   (b) *Memory-Retrieval:* Algorithms must dynamically retrieve appropriate memory samples by considering (a) relevant current queries, (b) the QA's confidence in those queries, and (c) the confidence of the responding RAs. Algorithms can also consider the regularization effect of noisy responses in selecting the most pertinent memory samples to improve performance.

   (c) *Adaptive-Learning:* Algorithms must modulate the importance of potentially incorrect responses based on their confidence. These algorithms may also consider the regularization effect of noisy responses to improve performance, balancing stability and regularization as a trade-off.

   Our proposed *Memory-Update* method improves QA's local performance on all lifelong learned tasks by 44.17% on CIFAR-100 and 26.8% on MiniImageNet (see Section 4.3).

**We present the conceptual (i) and technical contributions (ii, iii) of this paper below:**

**(i)** We introduce PEEPLL, a framework for solving distributed Multi-Agent Lifelong Learning, addressing the unique challenges of integrating multi-agent coordination with lifelong learning.

**(ii)** We set initial benchmarks for the components of PEEPLL. Within our solution to PEEPLL, an agent with no environmental supervision achieves higher lifelong performance than an LL agent with complete environmental supervision while minimizing communication overhead.

**(iii)** We introduce an application of a Variational Classifier for superior uncertainty quantification compared to Entropy-based methods.

## 2 Related Work

Lifelong Learning (LL) seeks to enable AI systems to continually improve their capabilities through experience beyond the limited-time training phase. Old versions of LL described systems that updated data structures while leaving the underlying neural networks unchanged (Thrun & Mitchell, 1995; Thrun, 1998; Ruvolo & Eaton, 2013). Following DARPA's L2M program, modern LL emphasizes continual updates to the neural network itself, i.e., the underlying computational AI model. This paradigm shift is supported by the Super-Turing computational theory (Siegelmann, 1995; 1998), an alternative to the Turing computing model. Unlike Turing computation, which relies on a fixed program to process input, Super-Turing, like biological systems, adapts its program based on input; this is the basis for LL. Due to its ongoing adaptability, LL, like Super-Turing, is computationally more powerful and produces richer output behaviors than the fixed Turing model or traditional neural networks. PEEPLL extends these principles to a distributed multi-agent setup to make LL more practical and scalable by reducing reliance on environmental supervision.

We next discuss Single-Agent Lifelong Learning, given the literature's gap in Multi-Agent Online Lifelong Learning, and Multi-Agent Lifelong Learning for offline implementations. Finally, we include the rationale behind the choice of a fully distributed setup in our study.

**Single-Agent Lifelong Learning:** Lifelong learning strategies can be classified into three categories: Memory-Based, Architecture-Based, and Regularization-Based Methods. We focus on Memory-Based Lifelong Learning strategies within the Class-Incremental setting, where data of unknown classes are presented sequentially in distinct, non-overlapping 'tasks' to the lifelong learning agent. Architecture-based methods are incompatible with this setting (Mai et al., 2021); hence, we do not focus on these. Regularization-based methods such as Elastic Weight Consolidation (EWC) (Kirkpatrick et al., 2016) struggle with this setting (Lesort et al., 2021). EWC prevents forgetting by penalizing changes to parameters important for previously learned tasks, using a Fisher information matrix to measure importance. Memory-based lifelong learning methods mitigate forgetting by storing and replaying (relearning) data from previously seen classes. This typically involves *Memory-Update* selecting what to store and *Memory-Retrieval* selecting what to replay. Mai et al. (2020) use Shapley values for informed retrieval; Aljundi et al. (2019a) retrieve samples that maximize learning interference; Aljundi et al. (2019b) focus on gradient diversity while storing in the memory; Chaudhry et al. (2018b) constraint the model's loss on its memory samples to mitigate forgetting; Prabhu et al. (2020) (GDumb) adopt a simple, balanced sampling approach, retraining the network from scratch at test time. Despite its simplicity, GDumb is more effective than many strategies; however, it is an offline strategy. Finally, Experience Replay (ER in Table 2) (Chaudhry et al., 2019) uses random sampling for memory management (update and retrieval) and often surpasses other complex methods. ER is considered state-of-the-art for Single-Agent Online Lifelong Learning (van de Ven et al., 2022). Naive strategies fine-tune the model on incoming data without replaying previously learned data. IID strategies sample independently and identically from all of the training data.

**Multi-Agent Lifelong Learning:** In Rostami et al. (2017), each agent knows specific distributions of the dataset. They assume that this distribution is fixed and known. Their lifelong learning agent is designed to be aware of all classes and learns only from out-of-distribution data, known as the Domain-Incremental Scenario. Our work differs in two key aspects: Firstly, we do not predefine or know how tasks are distributed among agents, as this is not a fair assumption for lifelong learning agents. Secondly, our agents are strictly unaware of certain classes, focusing instead on Class-Incremental Lifelong Learning. Babakniya et al. (2023) tackle catastrophic forgetting in Federated Learning (FL) using a global generative model for data-free knowledge distillation, focusing on privacy but increasing communication overhead and presenting a risk of single point of failure due to server centrality (centralized topology of communication). Instead, our work focuses on a Peer-to-Peer topology.

**Multi-Agent Architecture :** After thoroughly evaluating communication topologies (Verbraeken et al., 2019), we selected a fully distributed peer-to-peer architecture for the PEEPLL system. Although centralized systems offer quick synchronization, they struggle with scalability and single-node failure risks; decentralized systems improve scalability but still have communication overhead and node failure risks. Peer-to-peer networks avoid these issues but face challenges in synchronizing agent knowledge. PEEPLL addresses this through Confidence-Evaluation methods for knowledge alignment and Lifelong Learning strategies for continuous knowledge synchronization.

We compare our solutions to PEEPLL with ER, EWC, GDumb, Naive, and IID. We implement these with our modified data introduction regime (Section 4). Further, we propose that GDumb serves as a multi-agent LL baseline as well. This is because if we assume 100% accurate communication and that all agents in the network share a common memory, an agent can retrain its network using all relevant memory samples when encountering new classes. Although this is not a fully distributed setup, it is a strong baseline to beat.

## 3 PEEPLL: Our Approach

This section presents our solutions to the components of the PEEPLL mechanism discussed in Section 1.1 - (1) Confidence-Evaluation Strategy, (2) Selective Response Filtering, and (3) Lifelong Learning in PEEPLL.

### 3.1 Confidence-Evaluation Strategy

We introduce a modified compute-efficient version of Variational Autoencoders for image classification without input reconstruction to quantify uncertainty in model prediction. This architecture is similar to the re-

cently proposed Variational Classifier (VC) by Dhuliawala et al. (2024). Since our study focuses on Memory-Based Lifelong Learning, where a subset of experiences is stored in the agent's memory, our uncertainty quantification method is targeted for such agents.

We first discuss our Variational Classifier's *Architecture and Training*. Then, detail how the model's uncertainty in prediction (inversely, its confidence in prediction) is discerned.

*Architecture and Training:* VC's architecture comprises two primary components:
1. **Encoder:** Takes the image as input ('x') and outputs $z_{\text{mean}}(x)$ and $z_{\text{log\_var}}(x)$. $z_{\text{mean}}(x)$ and $z_{\text{log\_var}}(x)$ are used to define the probability distribution from which a latent code $z$ for the image is sampled.

The Encoder is optimized using a Kullback–Leibler divergence loss to facilitate a continuous latent space representation. The Kullback–Leibler divergence loss can be mathematically expressed as:

$$\text{KL}(q_\phi(z|x)||p(z)) = -\frac{1}{2}\sum_{k=1}^{K}\left(1 + z_{\text{log\_var}}^k(x) - (z_{\text{mean}}^k(x))^2 - e^{z_{\text{log\_var}}^k(x)}\right)$$

where $q_\phi(\mathbf{z}|\mathbf{x})$ is the approximate posterior distribution of z given x, parameterized by $\phi$ (encoder), $\mathbf{p(z)}$ is the prior distribution over z (Gaussian), $\mathbf{K}$ is the dimension of $z_{mean}$ and $z_{log\_var}$, $\mathbf{z}_{\text{mean}}^k$ is $k$-th entry of the $z_{mean}$ vector, and $\mathbf{z}_{\text{log\_var}}^k$ is $k$-th entry of the $z_{log\_var}$ vector.

2. **Decoder:** Takes in the latent code $z$ as input and outputs a classification layer. This component is trained using a Cross-Entropy loss to predict the class of the input image. Note here that we forgo input reconstruction to conserve computational resources; hence, no reconstruction loss is used for training.

*Confidence-Evaluation mechanism*: Given a query $q$, the encoder evaluates $z_{\text{mean}}^q$ and $z_{\text{log\_var}}^q$, and a latent code $z$ is sampled. The decoder then takes in $z$ and yields a prediction $p_q$ for $q$. If there are associated memory samples for the label $p_q$, the corresponding $z_{\text{mean}}^{\text{memory}}$ and $z_{\text{log\_var}}^{\text{memory}}$ for each memory sample is retrieved. Using these, we evaluate (1) *Semantic Disparity*, (2) *Dispersion Disparity*, and (3) Entropy, which are then averaged to produce the confidence score.

(1) *Semantic Disparity* captures the disparity in the property of the mean - position of the query in the latent space versus the mean position of the model's memory of the predicted label for the query. We define *Semantic Disparity* using Euclidean distance:

$$d_{\text{semantic}}(q, \text{memory}) = ||z_{\text{mean}}^q - \overline{z_{\text{mean}}^{\text{memory}}}||$$

where $\overline{z_{\text{mean}}^{\text{memory}}}$ represents $\frac{1}{N}\sum_i z_{\text{memory}^i}^{\text{memory}}$ where N is the total number of retrieved memory samples. $z_{\text{mean}}$ represents the mean of the latent variable's probability distribution for a given input; it is where the model predicts the center of the input's position in the latent space would be.

(2) *Dispersion Disparity* captures the disparity in the property of variance - the variance of the query versus the variance of the model's memory of the predicted label for the query. We define *Dispersion Disparity* using Manhattan distance:

$$d_{\text{dispersion}}(q, \text{memory}) = ||\exp(z_{\text{log\_var}}^q) - \overline{\exp(z_{\text{log\_var}}^{\text{memory}})}||_1$$

where $\overline{z_{\text{log\_var}}^{\text{memory}}}$ represents $\frac{1}{N}\sum_i z_{\text{log\_var}}^{\text{memory}^i}$ where N is the number of retrieved memory samples. $z_{\text{log\_var}}$ is the logarithm of the variance of the distribution of the latent variable. Intuitively, the variance represents the spread or dispersion of the latent representations around $z_{\text{mean}}$. A larger variance (higher $\exp(z_{\text{log\_var}})$) indicates the model's higher uncertainty in pinpointing the class for similar images.

The choice of Euclidean distance for the *Semantic Distance* is to measure the spatial distance between the query's *z_mean* and the memory samples' *z_mean* in the latent space, as *z_mean* represents a meaningful 'physical' location in this space. In contrast, *z_var* does not correspond to a physical spatial location but rather indicates the variance across dimensions. Therefore, we use the Manhattan Distance to evaluate the sum of disparities between the query's and memory samples' variance across all dimensions. After

experimenting with various distance measures, we find our two chosen distance measures to be the most effective.

*Semantic & Dispersion Disparity* are transformed into confidence scores via a negative exponential function, $\exp(-\alpha \cdot d_{\text{semantic}})$ (denoted $C_{\text{semantic}}$), $\exp(-\beta \cdot d_{\text{dispersion}})$ (denoted $C_{\text{dispersion}}$). This ensures that the scores are bounded between $[0, 1]$ and are inversely proportional to the assessed distances. $\alpha$ and $\beta$ are scaling factors. The transformed distances are subsequently normalized (denoted $\hat{C}_{\text{semantics}}$, and $\hat{C}_{\text{dispersion}}$ respectively).

(3) Entropy (Shannon, 1948) is defined as $H(\mathbf{p}) = -\sum p_i \cdot \log_2(p_i + \epsilon)$, where $\mathbf{p}$ is the probability distribution of the agent's predictions, and $\epsilon$ (a small number) prevents undefined logarithmic operations.

Finally, the mean of the three is calculated to ensure that the resultant Triplet Uncertainty Evaluation (TRUE) confidence score falls within the range $[0, 1]$:

$$\text{TRUE} = \frac{\hat{C}_{\text{semantics}} + \hat{C}_{\text{dispersion}} + \text{Entropy}}{3}$$

Refer to Figure 12 and Algorithm 2 in the Appendix for a visualization and a pseudocode to evaluate TRUE.

The QA uses TRUE to decide whether to initiate communication with the RAs, and the RAs use TRUE to determine whether their responses should be transmitted back to the QA. We measure TRUE's effectiveness by the 'Sharing Accuracy' metric: the proportion of responses sent by RAs to the QA that were correct. We demonstrate that TRUE offers a higher Sharing Accuracy than Entropy-based measures. This implies that it is possible to generate meaningful latent representations despite forgoing reconstruction.

### 3.2   Selective Response Filtering

We propose three approaches to filter out pertinent responses at the QA once the RAs have responded.

1. Majority Voting: Response agents are grouped by the label they are predicting. The label with the highest number of agent endorsements is selected.

2. Most-Confident-Group (MCG): Response agents are grouped by their predicted labels. The group with the highest average confidence is selected.

3. Intelligent Comparative Filtering (ICF): The responses of RAs that predict the label $l$ are rejected if any other RA that has memory samples of the label $l$ does not predict class $l$.

The effectiveness of a filter lies in accurately identifying and accepting the correct responses (increasing the proportion of correct responses in the accepted subset).

### 3.3   Lifelong Learning & *Memory-Update*

As the QA receives responses from RAs, it stores these responses and their corresponding confidence levels in a memory organized by class. We allocate a budget of 50 samples per class. To dynamize our *Memory-Update* mechanism, once the memory reaches 50 samples for a class, it dynamically replaces the least confident samples with new, higher-confidence responses.

## 4   Results & Analysis

We start this section outlining the modified traditional LL experimental setup to fairly evaluate solutions to PEEPLL's components and present the results in Sections 4.1-4.3.

**Data Distribution and Agent Roles:** The data in the training set is divided into two parts: (1.1) pretraining data for all agents and (1.2) lifelong learning data for the QA. The pretraining data (1.1) is further divided into (1.1.Q-Pre) and (1.1.R-Pre). (1.1.Q-Pre) contains data from 5 classes assigned for the QA's

pretraining. (1.1.R-Pre) contains data from classes assigned for the pretraining of the RAs. These classes are distributed among RAs, ensuring that 2-3 RAs are assigned to each class to foster diverse responses. Each RA is trained on data from its assigned classes in (1.1.R-Pre). From the lifelong learning data (1.2), the data for classes not in (1.1.Q-Pre) are extracted and referred to as (1.2.LL). (1.2.LL) is segmented into distinct 'tasks,' each comprising a unique set of classes to facilitate the Class-Incremental Scenario as described by Aljundi et al. (2019b;a); Mai et al. (2020). $\text{Task}_t$ has classes, $\mathcal{C}_t$, absent in previously seen tasks, $\text{Task}_{1-(t-1)}$, such that $\mathcal{C}_t \subset \mathcal{C} \setminus \mathcal{C}_{1-(t-1)}$. The QA is incrementally introduced to a $\text{Task}_t$, which it learns in communication with the other agents (RAs) in the network. Note here that since we differentiated the data into the pretraining data (1.1) and the lifelong learning data (1.2), we can fairly evaluate the RA's responses and the confidence-evaluation strategy that discerns the RA's confidence while responding. In evaluation, we follow the single-head evaluation setup (Chaudhry et al., 2018a) where the QA does not know the task it is introduced, so it has to choose between all possible classes for classification.

Our experimental setup considers 20 agents - 1 QA and 19 RAs - that are pre-trained on subsets of 5-10 classes each, achieving 75-85% accuracy on their specific tasks. This simulates scenarios with imperfect, non-teacher-like agents. We evaluate our solutions on vision datasets CIFAR-100 and MiniImageNet.

### 4.1 Confidence-Evaluation

**Benchmarks:** First, we establish benchmarks for comparing confidence-evaluation strategies.
(1) *Baseline:* Our baseline strategy for assessing agent confidence employs softmax probabilities defined as $\sigma(\mathbf{p})_i = \frac{e^{p_i}}{\sum_{j=1}^{K} e^{p_j}}$. This transforms the output of the model - vector $\mathbf{p}$ - into a probability distribution. This distribution provides insight into the agent's confidence level regarding its response. We include results for this in our Appendix (Figure 13), given its documented inferior performance relative to entropy-based measures (Yona et al., 2022), which constitute the primary focus of our comparative analysis and discussion.
(2) *Entropy-Based Measure:* To enhance our confidence assessment, we use Entropy, introduced by Shannon (1948) and a widely effective and accepted metric (Yona et al., 2022; Chen et al., 2021) for uncertainty quantification. It quantifies 'chaos' in a probability distribution. Since Entropy $\geq 0$, we transform this entropy into a confidence score using a negative exponential mapping, $e^{-H(\mathbf{p})}$. This transformation maps higher entropy (chaos) to lower confidence and vice versa. Note that this transformation clips the confidence score from 0 to 1.

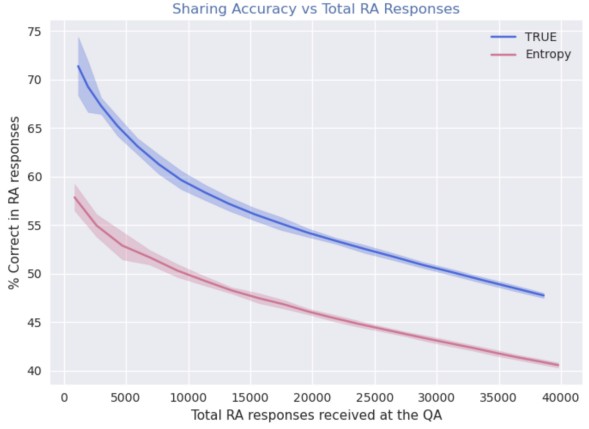
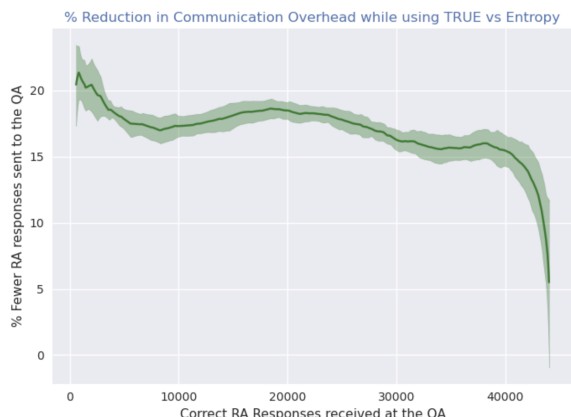

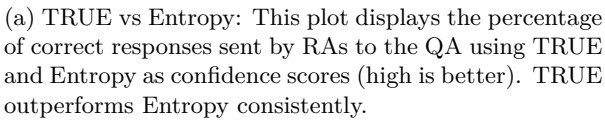

(a) TRUE vs Entropy: This plot displays the percentage of correct responses sent by RAs to the QA using TRUE and Entropy as confidence scores (high is better). TRUE outperforms Entropy consistently.

(b) A PEEPLL system using TRUE needs 5-20% fewer total RA responses (vs. Entropy) for the QA to receive the same number of correct responses. This contributes to a reduction in system communication overhead.

Figure 2: These plots correspond to evaluation on CIFAR-100. See Figure 10 for MiniImageNet in Appendix.

**TRUE Results:** The RAs compare their confidence against a threshold, and if their confidence is greater than the threshold, they send their responses back to the QA. We measure *Sharing Accuracy*: the percentage

of responses sent by RAs to the QAs that match the ground truth. For standardized analysis, in this section, we report numbers that refer to the thresholds for a 1:1 Sharing Ratio - the number of responses the QA receives is close to the number of queries it sends. See Figure 2a for a comparison of Entropy and TRUE across thresholds. TRUE shows a 19.29% improvement in Sharing Accuracy (from 46.3% to 55.2%) on CIFAR-100 and a 7.03% improvement (from 68.4% to 73.2%) on MiniImageNet over using Entropy. An ablation experiment demonstrates that incorporating the Semantic Distance ($d_{\text{semantic}}$) yields an 11.7% improvement (47% to 52%) and the Dispersion Distance ($d_{\text{dispersion}}$) yields a 16.75% improvement (46.5% to 54.3%) in Sharing Accuracy on CIFAR-100. The efficacy of TRUE also demonstrates that it is possible to derive a meaningful latent space in Variational Autoencoders without reconstruction.

Since the TRUE confidence score aligns more closely to response correctness than Entropy, the RAs are better equipped to transmit responses judiciously. As a result, it helps restrict communication. To get 'x' correct responses at the QA, the RAs need to send 5-20% fewer responses on CIFAR100 (Figure 2b) and 5-15% on MiniImageNet (Figure 10b in Appendix) while using TRUE vs Entropy. This is a critical analysis to assess the reduction in communication overhead.

We find that knowledge sharing among agents scales inversely with the number of agents in the environment (see Figure 7, Appendix). This trend is expected as the likelihood of randomly choosing the correct agent to listen to decreases with an increasing number of agents, making the task progressively harder. However, our results indicate that TRUE consistently outperforms Entropy across varying numbers of agents.

The computational complexity for an agent in PEEPLL, in terms of the scalable symbols - Q total queries, N(., out) the number of RAs a QA contacts for a query, and K total classes - is $\mathbf{O(Q \times N(., out) \times K)}$. Refer to Section E of the Appendix for a derivation of the complexity.

While TRUE has demonstrated clear advantages over entropy-based methods for uncertainty quantification, we note that its reliance on a variational classifier limits its direct application to pre-trained models and purely discriminative architectures. This presents interesting opportunities to advance and expand the applicability of TRUE to such models by modifying the architecture and fine-tuning.

## 4.2 Selective Response Filter:

The QA accepts a subset of responses from all received RA responses. We measure the percentage of correct responses in the accepted subset of responses. We provide a structured display of results in Table 1.

Table 1: Efficacy of Filters applied at the QA

| Method | Sharing Accuracy (%) | % Improvement (↑) | Sharing Accuracy (%) | % Improvement (↑) |
|---|---|---|---|---|
| | CIFAR-100 | | MiniImageNet | |
| TRUE | 55.2 | - | 73.17 | - |
| TRUE + Majority | 66.2 | 19.92 | 83.56 | 14.02 |
| TRUE + MCG | 65.8 | 19.13 | 82.04 | 12.14 |
| TRUE + ICF | 82.8 | 50 | 91.22 | 24.67 |
| TRUE + Majority + ICF | 83.3 | 50.9 | 91.91 | 25.6 |
| **TRUE + REFINE** | **83.9** | **51.99** | **92.04** | **25.79** |

The QA employs our proposed filters to select a subset of responses received from all RAs. This table displays the percentage of accepted responses that match the ground truth (Sharing Accuracy). REFINE (ICF with MCG) is the most effective filter. These values correspond to a 1:1 Sharing Ratio (thresholds where the number of accepted responses by the QA is close to the number of queries sent out by the QA). For an equivalent analysis across thresholds, see Figure 3.

As seen in Figure 3b (and 11b in Appendix), the Most-Confident-Group (MCG) approach exhibits enhanced effectiveness at lower thresholds (more RAs respond), suggesting that in scenarios where response criteria are

less strict, prioritizing collective agent confidence yields better results than majority voting. The rationale is that when agents with low confidence are allowed to respond, they cannot be taken at face value, and a further focus on their confidence is required. As the threshold increases, the Majority-Voting method parallels and occasionally surpasses the MCG approach. This trend indicates that with higher confidence thresholds, agents become more reliable, allowing responses with the most agent endorsements to be selected with greater assurance of accuracy. Implementing Intelligent Comparative Filtering (ICF) with the MCG filter retains a consistent edge over Majority Voting with ICF.

ICF consistently outperforms non-ICF methods. ICF with MCG constitutes our most effective filter, which we term "**REFINE**" filter; see Table 1 and Figure 3.

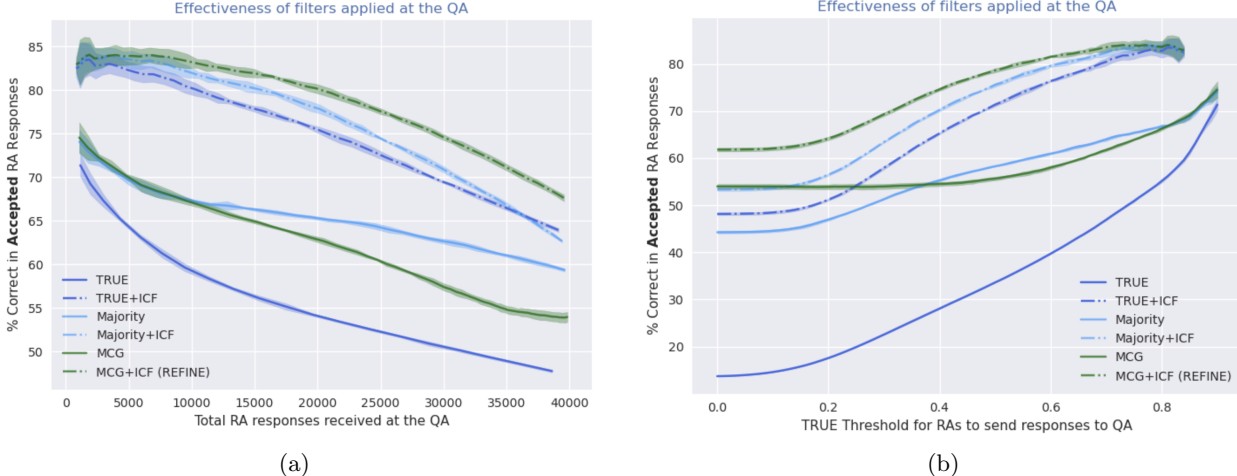

(a)                                                     (b)

Figure 3: The QA applies our proposed filters to select a subset of responses received from RAs. These plots display the percentage of correct responses in the accepted subset (filter efficacy) across different confidence thresholds that the RAs must meet before sending their responses to the QA. Both plots are equivalent: the left plot's x-axis (total responses the QA receives) is controlled by the confidence threshold parameter shown on the right plot's x-axis. In the left figure, we cut off at a threshold of a 1:2 ratio of Queries asked to Responses received for clarity. Our REFINE filter consistently outperforms all other filters. This plot corresponds to the evaluation on CIFAR-100 (for MiniImageNet, see Appendix Figure 11).

### 4.3 Lifelong Learning

For the sake of initial comparison, we designate one agent as the QA, while the remaining agents act as RAs. After the QA learns all Tasks$_{1-T}$ incrementally in communication with its imperfect peers, we measure its local performance on the complete test set. For standardized testing, we evaluate the QA's performance by choosing the communication strategy, which includes the thresholds for TRUE and filter choice, such that there is a 1:1 Sharing Ratio, that is, the number of peer responses the QA accepts and learns from is close to the number of queries it seeks assistance on.

We compare our solution to PEEPLL against a Single-Agent that learns with complete supervision - the QA learns new tasks using ground truth labels, same as the Experience Replay (ER) (Chaudhry et al., 2019) strategy. This refers to 'ER', as seen in Table 2. ER has been consistently shown to outperform other LL strategies and is state-of-the-art (van de Ven et al., 2022). Matching ER necessitates advanced strategies in Confidence-Evaluation, Selective Response Filter, and Lifelong Learning in PEEPLL, particularly those that judiciously utilize potentially incorrect responses.

As shown in Table 2, despite learning from noisy peer responses, our solutions to PEEPLL outperform ER, the Supervised Single-Agent, which relies on ground-truth responses (complete environmental supervision), by 2.1% on CIFAR-100 and 2.1% on MiniImageNet.

Table 2: The QA's local performance on the complete test set after lifelong learning of all tasks.

| Communication Strategy | Accuracy (%) | Quality (%) | Memory Samples | Accuracy (%) | Quality (%) | Memory Samples |
|---|---|---|---|---|---|---|
| | CIFAR-100 | | | MiniImageNet | | |
| Offline Single-Agent (10 epochs) | | | | | | |
| IID | 58.2 ± 1.1 | 100 | - | 52.4 ± 1.2 | 100 | - |
| Naive (fine-tune) | 4.9 ± 0.2 | 100 | - | 4.6 ± 0.2 | 100 | - |
| EWC | 3.9 ± 0.3 | 100 | - | 3.1 ± 0.2 | 100 | - |
| GDumb | **39.5 ± 0.8** | **100** | **5k** | **47 ± 1.3** | **100** | **5k** |
| Offline PEEPLL (10 epochs) | | | | | | |
| PEEPLL+REFINE - **Naive** | 4.2 ± 0.3 | 83.9 | - | 4.1 ± 0.3 | 92.04 | - |
| Online Single-Agent | | | | | | |
| IID | 36.7 ± 0.9 | 100 | - | 34.5 ± 0.8 | 100 | - |
| Naive | 2.4 ± 0.1 | 100 | - | 2.2 ± 0.2 | 100 | - |
| GDumb | 3.6 ± 0.5 | 100 | 5k | 9.2 ± 0.7 | 100 | 5k |
| ER | 29.2 ± 0.5 | 100 | 5k | 24.8 ± 0.47 | 100 | 5k |
| Online PEEPLL | | | | | | |
| PEEPLL+REFINE - **Naive** | 2.1 ± 0.1 | 83.9 | - | 1.9 ± 0.1 | 92.04 | - |
| Online PEEPLL w. Confidence | | | | | | |
| Entropy | 26.5 ± 0.4 | 46.3 | 5k | 22.35 ± 0.3 | 68.4 | 5k |
| TRUE | 29.7 ± 0.15 | 55.2 | 4.5k | 25.6 ± 0.58 | 73.17 | 4.3k |
| Online PEEPLL w. Confidence + Filter | | | | | | |
| TRUE + Majority | 30.8 ± 0.57 | 66.2 | 4.7k | **26.9 ± 0.76** | **83.56** | **4.6k** |
| TRUE + MCG | 28.9 ± 0.39 | 65.8 | 4.9k | 26.7 ± 0.76 | 82.05 | 5k |
| TRUE + ICF | 31.1 ± 0.31 | 82.8 | 4.5k | 26.9 ± 0.92 | 91.22 | 4.3k |
| TRUE + Majority + ICF | 30.6 ± 0.27 | 83.3 | 4.5k | 25.6 ± 0.49 | 91.91 | 4.7k |
| TRUE + REFINE | **31.3 ± 0.72** | **83.9** | **4.5k** | 26.1 ± 1.10 | 92.04 | 4.7k |

This table presents the QA's local performance (Accuracy (%)) on the complete test set following lifelong learning of all tasks. Other metrics shown include Quality (%) and Memory Samples. *Quality (%)* refers to the percentage of correct responses the QA learns from, determined by the communication strategy's Sharing Accuracy. Single-Agents learn from the environment with 100% Quality. PEEPLL agents may not fill their memory buffers if they do not receive enough RA responses for certain classes, while Naive strategies do not utilize Replay/Memory. Note that although GDumb is not specifically designed for Lifelong Learning, it is used as a strong baseline for comparison. Our results demonstrate that our PEEPLL agent outperforms ER (the state-of-the-art Online Single-Agent LL) despite learning from noisy peer responses.

We find that the noisy responses from which the QA learns, in fact, generate a beneficial regularization effect in its learning process. To analyze this effect and the QA's ability to effectively learn from potentially incorrect peer responses, we monitor its performance on future yet-to-be-introduced tasks throughout its lifelong learning journey (see Figures 5a and 8a in Appendix). Observe that a PEEPLL system using a communication strategy with lower Sharing Accuracy shows better QA performance on future tasks. This suggests that responses deemed 'incorrect' may still hold relevance for queries of another class with similar characteristics. This is also supported by Figure 6a, where the QA's initial confidence in each subsequent task progressively increases throughout its learning journey. This is because the QA has previously, albeit briefly, learned about these classes from 'incorrect' responses. We hypothesize that briefly learning about future classes while learning similar classes beforehand produces a regularization effect that aids in more effective learning of those future classes when they become relevant. This is supported by the trends in Figure 5b (and 8b, Appendix), where initially, the Single-Agent (ER) and the QA learning from peer responses perform comparably, but the QA outperforms in later stages. This phenomenon is akin to and is further supported by the less challenging Domain-Incremental Learning (Domain-IL) scenario in lifelong learning, where an agent incrementally learns the same classes from different data distributions. Research has shown that LL agents find learning easier in Domain-IL than in Class-IL (van de Ven et al., 2022), explaining the effective learning of QA despite potentially incorrect responses from peers. However, note that the QA's improved performance

on future tasks comes at the expense of current task performance. This trade-off has implications for the LL algorithms and communication strategies that will be developed for PEEPLL agents.

Note that the communication strategy influences the subset of responses chosen for learning and the percentage of correct responses within that subset, affecting the regularization effect and the QA's learning for current and future tasks. While we demonstrate that a regularization effect exists, future work will examine in detail the impact of our proposed communication strategies on the regularization effect and the QA's performance on current and future tasks.

Another important observation from Figures 5a (and 8a in Appendix) is that the QA performs better on future tasks within the PEEPLL system that uses TRUE as the communication strategy compared to Entropy, despite TRUE having higher Sharing Accuracy. This indicates that the TRUE score not only aligns more closely with response correctness but also more effectively captures query similarity. In other words, a high TRUE score for an incorrect response suggests that the predicted class is likely similar to the query's actual class. This feature is weaker with entropy; a high entropy score for an incorrect response indicates that the predicted class is less likely, compared to TRUE, to be similar to the query's original class.

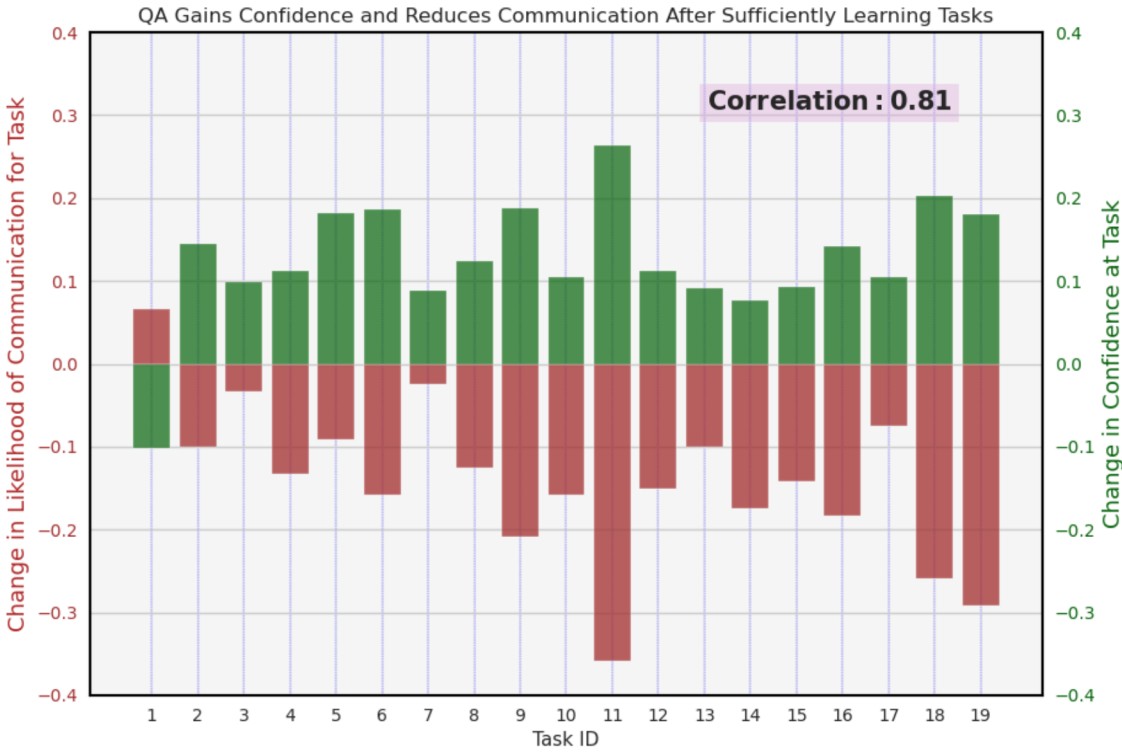

Figure 4: As the QA learns each task incrementally (x-axis), it gains confidence in answering queries related to that task (green) and consequently reduces communication calls (maroon) to the RAs for assistance. A green bar indicates the change in the QA's confidence in answering queries from the start to the end of each task. A maroon bar shows the change in the number of times the QA initiates communication with RAs per query from the start to the end of each task. The correlation between the change in QA's TRUE confidence and the reduction in communication calls is 0.81, with a p-value of 2.98e-05. This plot corresponds to a PEEPLL agent learning MiniImageNet with TRUE: Table 2, Row 12, and performs 25.6% on the test set.

Figure 4 illustrates how the PEEPLL framework reduces system communication overhead over time. As the QA learns a task, its confidence in answering the task's queries increases. And since the QA only requests assistance when it is underconfident in answering a query, the increase in confidence leads to fewer requests for assistance. In Figure 4, the change in QA's confidence in each task is shown in green, while the change in the likelihood of initiating a communication call is shown in maroon. We find that the correlation between the change in QA's confidence and the change in communication likelihood is strong, 0.81, with a

p-value of 2.98e-05, meaning it is statistically significant. Figure 6 in the Supplementary Material shows the unprocessed data of growing confidence and a decrease in communication. This reduction in communication overhead helps make multi-agent systems practical.

Note that even after the QA completes its subsequent lifelong learning tasks, the TRUE score continues to exhibit expected trends with each new task - initially, confidence is low, then it increases over time (see Figures 4 and 6). This demonstrates TRUE's context-aware adaptability to the QA's continual learning capability, which is crucial for lifelong learning applications.

Our Dynamic Memory-Update mechanism improves QA's performance by 44.17% on CIFAR-100's test set (21.71% to 31.3% on CIFAR-100 with PEEPLL using TRUE + REFINE) and 26.8% on MiniImageNet's test set (21.21% to 26.9% on CIFAR-100 with PEEPLL using TRUE + Majority) compared to a static Memory-Update system that does not replace old, less-confident responses. This mechanism mitigates the adverse effects of previously received incorrect responses. This mechanism further supports the reality of multi-agent systems: if a QA requires assistance from a response agent that is skilled at a certain task yet hard to reach (e.g., offline or distant), the QA will now be able to receive the new responses once that agent becomes available and replace previous low-quality responses in memory. This stabilizes learning and improves the reliability of the system.

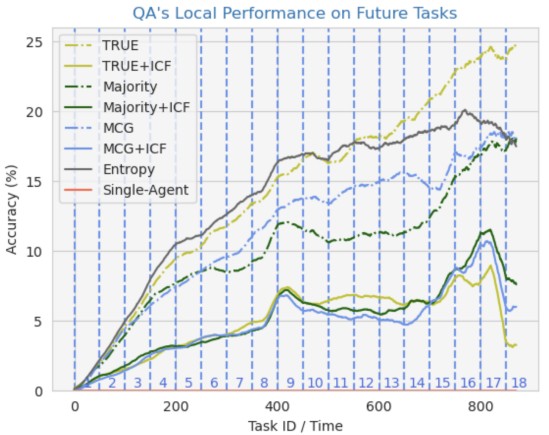

(a) QA's performance on yet-to-be-introduced tasks.

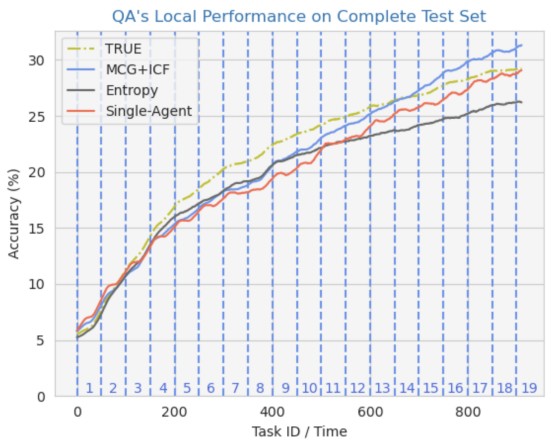

(b) QA's performance on all tasks.

Figure 5: Each vertical blue line marks the introduction of a new task to learn for the QA. In (a), observe that QA learning with communication strategy with lower sharing accuracy exhibits higher performance on untrained tasks. In (b), observe that QA learning from peer responses performs comparably to the single-agent initially but outperforms in the later stages. This plot corresponds to the evaluation on CIFAR-100. See Figure 8 in Appendix for MiniImageNet.

## 5    Conclusion

This paper introduces Peer Parallel Lifelong Learning (PEEPLL), the first distributed multi-agent lifelong learning framework. PEEPLL brings forth essential directions for realizing distributed lifelong learning, such as autonomously identifying novel tasks, navigating potentially incorrect responses from peers, and managing communication overhead. This paper lays the foundation for further research by setting initial benchmarks: TRUE for Confidence-Evaluation, REFINE for Selective Filtering, and Dynamic Memory-Update for Lifelong Learning in PEEPLL. Our results demonstrate that PEEPLL agents can outperform traditional LL agents with complete supervision, even when learning from potentially incorrect peer responses. By reducing reliance on environmental supervision, PEEPLL marks a step toward realizing seamless lifelong learning technologies at the edge. Furthermore, since PEEPLL's self-aware agents respond only when confident and seek peer assistance to learn only when underconfident, this minimizes risks during unprecedented events and reduces forgetting of previously learned knowledge caused by unnecessary learning while already confident. Thus, our

proposed PEEPLL mechanisms significantly improve the safety and adaptability of LL systems in dynamic learning conditions. Future research will investigate the mixed dependence on environmental supervision and peer assistance and explore variations in agent quantity, expertise, and memory budgets. We will also explore communication strategies with centralized support to develop more advanced communication strategies. These investigations will further refine PEEPLL's capabilities, paving the way for more seamless, resilient, and adaptive Lifelong Learning systems. Despite its controlled scope, this study establishes a foundation for research in distributed multi-agent lifelong learning. We encourage the research community to work toward this critical direction in AI, which holds significant implications for autonomous and semi-autonomous systems.

**Acknowledgments**

We sincerely thank the Air Force Research Laboratory (Distributed Lifelong Learning) and the DARPA ShELL (Shared Experience Lifelong Learning) program for supporting this research through their funding. We are also grateful to several students from the BINDS lab who helped at different stages of this project: Devdhar Patel suggested the cross-threshold TRUE evaluation incorporated in Figures 2a, 3, 10a, and 11. Adam Kohan contributed to the initial code, including the ER implementation, though the PEEPLL solution introduced in this paper diverged significantly from the preliminary efforts. Arjun Karuvally provided valuable feedback on writing, and Joshua Russell offered helpful suggestions for improving visual representations.

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

# A    Appendix

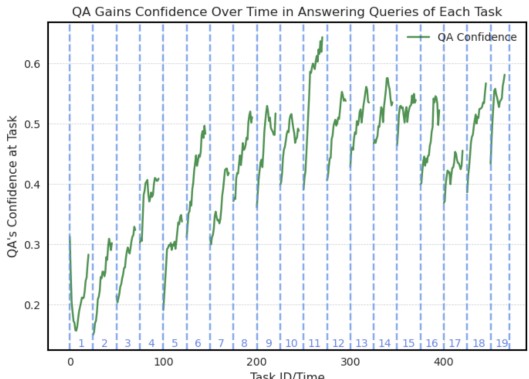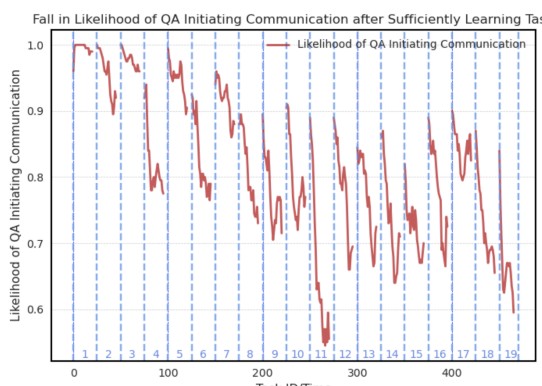

Figure 6: QA Gains Confidence & Reduces Communication: Each vertical blue line marks the introduction of a new task in the QA's lifelong learning journey on MiniImageNet. (a) Initially, the QA exhibits low confidence in queries related to the new task, but as it learns through communication with RAs, its confidence increases. (b) This increased confidence leads to self-reliance, reducing QA's number of calls to the network for responses and diminishing the system's communication overhead.

# B    Methodology

Here, we briefly discuss how we conduct experimental evaluations of our solutions to the different elements of the PEEPLL framework.

## B.1    Confidence-Evaluation Strategies:

*Setup:* During lifelong learning, the QA is introduced to queries from 1.3. The QA queries the RAs for inputs when their confidence is low. The RAs (pre-trained on 1.2) answer and evaluate their confidence in the received queries. Responses with confidence higher than a certain confidence threshold are sent back to the QA.

**Experimental Evaluation:**
1. We measure the proportion of responses that the RAs sent back to the QAs that matched the ground truth at various confidence thresholds (see Figure 2a).
2. We assess the reduction in total data sent from the RAs to the QAs to achieve the same number of correct responses compared to the entropy-based approach (see Figures 2b and 8). This metric is vital for reducing communication overhead in multi-agent settings.
3. We track the QA's growing confidence within specific tasks (Figure 4a). This helps us diminish communication needs as the QA becomes more adept at new tasks (Figure 4b). These demonstrations will showcase the confidence metric's adaptability to the QA's continual learning capability – an essential property in lifelong learning settings.
4. We also conduct a specific analysis at a 1:1 sharing ratio, where the number of responses received by the QA from the RAs equals the total number of queries processed (corresponding to the 1.3 data segment). This controlled setting provides a standardized basis for comparison under balanced data exchange conditions.

## B.2    Selective Response Filter:

*Setup:* Upon receipt of responses from RAs, the QA implements an additional selective response filter. This filtering mechanism leverages the broader response pool from multiple RAs, thereby facilitating more informed decision-making.

**Experimental Evaluation:**
The efficacy of the Selective Response Filter is measured by its ability to increase the proportion of the

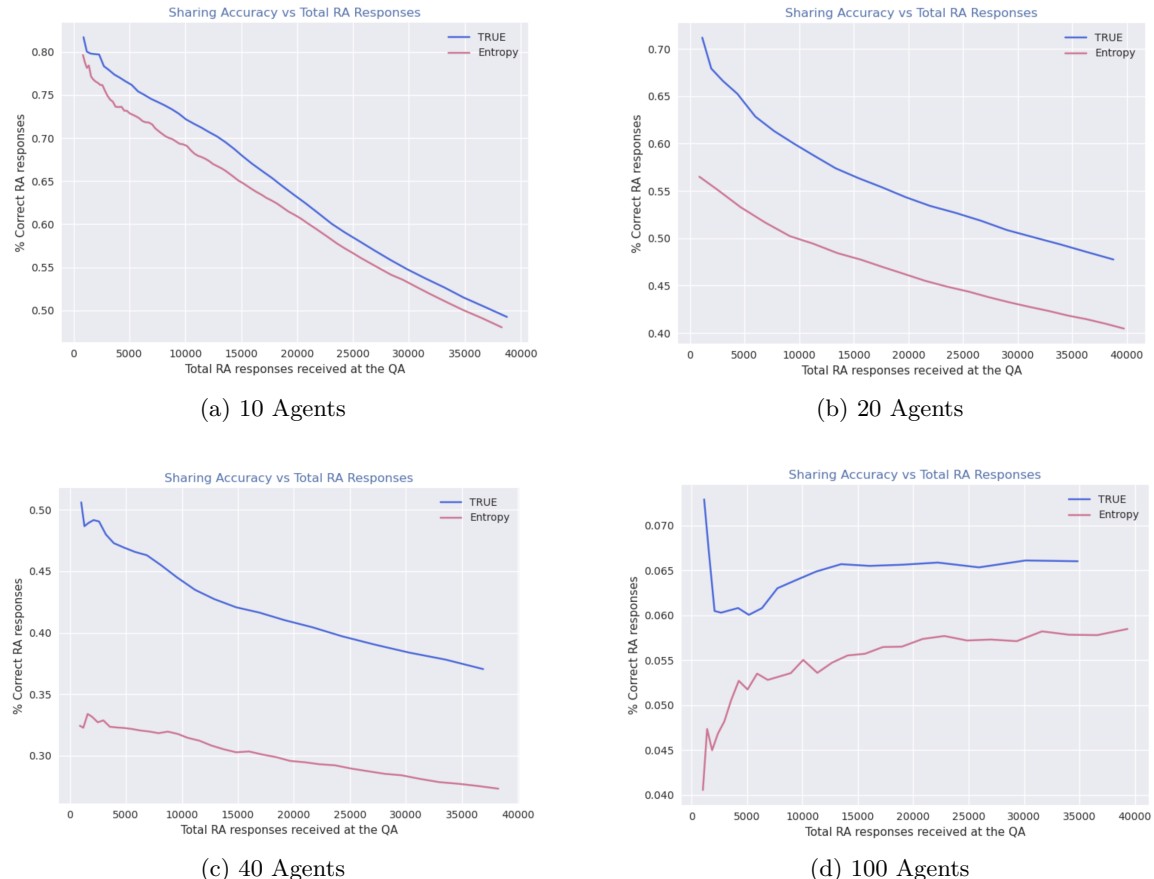

(a) 10 Agents

(b) 20 Agents

(c) 40 Agents

(d) 100 Agents

Figure 7: Sharing Accuracy Scaling with Number of Agents. As the number of agents increases, expectedly discerning the correct agent to listen to becomes increasingly challenging. Baseline probabilities for randomly selecting the correct agent are as follows: 0.1 for 10 agents, 0.05 for 20 agents, 0.025 for 40 agents, and 0.001 for 100 agents. For fair comparisons against these baselines, refer to when 20k RA responses are received at the QA, as the experiment considers 20k queries. While our current performance indicates progress, there remains significant room for improvement in developing more effective communication strategies for a higher number of agents. Notably, TRUE consistently outperforms Entropy, though their performance becomes nearly comparable with 100 agents.

correct responses in the accepted set (Figures 3 and 10, Table 1). For instance, if the filter receives 10,000 samples with 3,000 correct responses (proportion: 30%) and subsequently filters out 5,000 samples, retaining 2,500 correct responses (proportion: 50%), this indicates an increase in the proportion of correct responses (by 20%). Such analysis underscores the filter's capacity to effectively identify and preserve the most relevant and accurate responses. Moreover, we conduct specific 1:1 Sharing Analyses for these processes as well.

### B.3   Lifelong Learning:

*Setup:* After receiving responses from RAs, the QA further employs a selective filter to accept only the most pertinent answers. The QA then learns from this accepted set of responses.

**Experimental Evaluation:**

1. QA's Local Performance on the Complete Test: We assess the QA's final test performance when learning responses received using different communication strategies (Table 2 and Figure 6).

2. QA's Local Performance on Untrained Tasks: The QA's effectiveness on tasks it is not explicitly trained on is evaluated to understand the regularization effect, as discussed in Section 4.3, and the quality of 'bad'

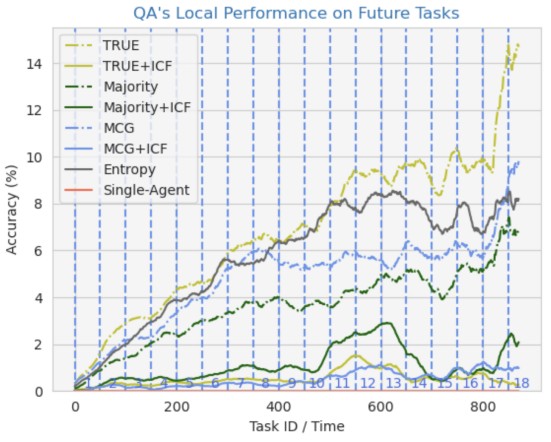
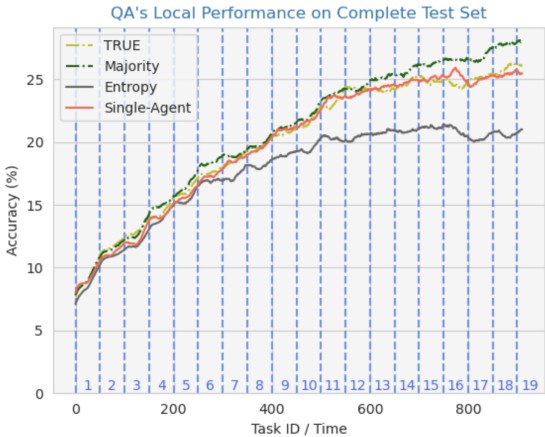

(a) QA's performance on yet-to-be-introduced tasks.

(b) QA's performance on all tasks.

Figure 8: Each vertical blue line marks the introduction of a new task to learn for the QA. In (a), observe that QA learning with communication strategy with lower sharing accuracy exhibits higher performance on untrained tasks. In (b), observe that QA learning from peer responses performs comparably to the single-agent initially but outperforms in the later stages. This plot corresponds to the evaluation on MiniImageNet.

---

**Algorithm 1** The PEEPLL Algorithm

---

1: **Input to QA:** Sample $x$, Maximum confidence $C_{\max}$, Confidence threshold $C_{\text{threshold}}$
2: At QA: Get QA's label on $x$, $l_x^{QA}$
3: At QA: Get QAs confidence on $x$, $c_x^{QA}$
4: At QA: If $c_x^{QA} < C_{\max}$, send $(x, c_x^{QA})$ to RAs
5: At RA: Get RA's label on $x$, $l_x^{RA}$
6: At RA: Get RA confidence on $x$, $c_x^{RA}$
7: At RA: If $c_x^{RA} > C_{\text{threshold}}$ and $c_x^{RA} > c_x^{QA}$, send $l_x^{RA}$ to QA
8: At QA: Collect all responses by RAs
9: At QA: Selectively filter which responses to accept
10: At QA: Update Memory with accepted responses
11: At QA: Learn Accepted Responses + Replay memory

---

responses (Figure 5).

## C   Implementation Details

We use VGG16 (Simonyan & Zisserman, 2015) as the backbone for our agents under PEEPLL. We deviate from the current lifelong learning research by doing so that currently uses ResNet18 (He et al., 2015). Our methodology utilizes VGG16 due to its lack of skip connections, enabling the isolation of our lifelong learning strategies' impact. ResNet18's skip connections, known to mitigate vanishing gradients, could introduce confounding factors.

PEEPLL models employ VGG16 as the encoder with a small MLP decoder for task-specific outputs. The total parameters of our PEEPLL model were 15417124.

We use the same Optimizer hyperparameters for pretraining and lifelong learning, as it would not be practical to tune those parameters further for online learning.

We implemented conventional single-agent LL strategies using Avalanche. We used maximum memory as 5k, the batch size of memory as 5k, and the ClassBalancedBuffer style of memory with no adaptive size. This is the same for our implementations of our PEEPLL strategies.

---

**Algorithm 2** Evaluating TRUE Confidence

---

1: **Input to Agent 'A':** Sample $x$
2: Get latent representations of $x$, $z^x_{\text{mean}}$, $z^x_{\text{logvar}}$
3: Prediction on $x$, $p_x$
4: Get label on $x$, $l_x = \text{argmax}(p_x)$
5: Get memory samples with label $l_x$, $\text{Memory}_{(x,l_x)}$
6: Get latent representation of retrieved memory samples, $\text{Memory}^{z_{\text{mean}}}_{(x,l_x)}$ and $\text{Memory}^{z_{\text{logvar}}}_{(x,l_x)}$
7: $d_{\text{semantic}} = ||z^x_{\text{mean}} - \text{Mean}(\text{Memory}^{z_{\text{mean}}}_{(x,l_x)})||$
8: $d_{\text{dispersion}} = ||z^x_{\text{logvar}} - \text{Mean}(\text{Memory}^{z_{\text{logvar}}}_{(x,l_x)})||_1$
9: $\text{entropy} = -\sum_i p_{x_i} \cdot \log_2(p_{x_i} + \epsilon)$
10: Transform into Confidence $C_{\text{semantic}} = e^{-d_{\text{semantic}}}$
11: Transform into Confidence $C_{\text{dispersion}} = e^{-d_{\text{dispersion}}}$
12: Transform into Confidence $C_{\text{entropy}} = e^{-d_{\text{entropy}}}$
13: Normalize scores
14: $\text{TRUE} = (C_{\text{semantic}} + C_{\text{dispersion}} + C_{\text{entropy}})/3$
15: **return** TRUE

---

# D Additional Results

Results for Comparison of our TRUE Confidence with Max-of-Softmax are illustrated in Figure 13.

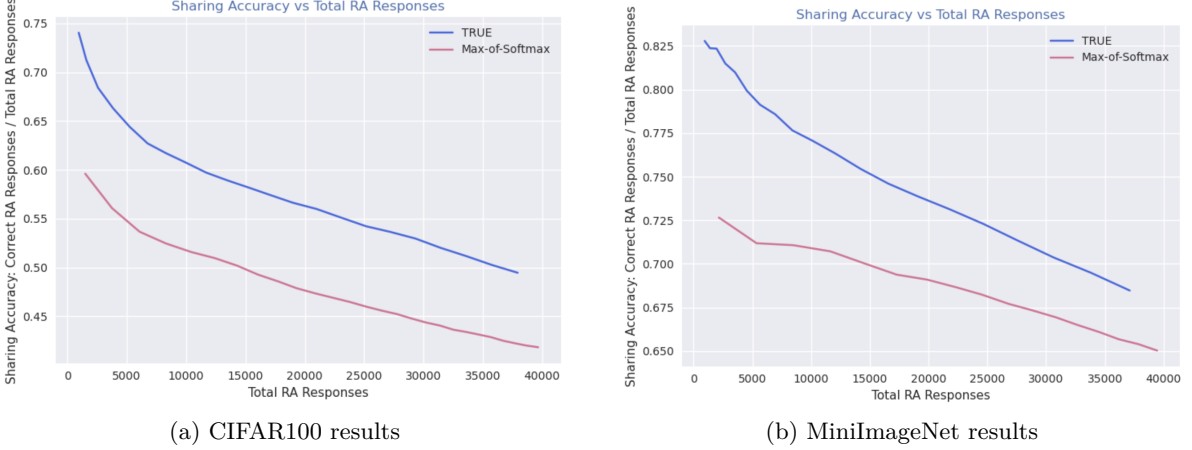

(a) CIFAR100 results          (b) MiniImageNet results

Figure 9: The TRUE score highly outperforms Max-of-Softmax as confidence. The plot shows the proportions of the total responses (from all RAs) that match the ground truth (higher is better).

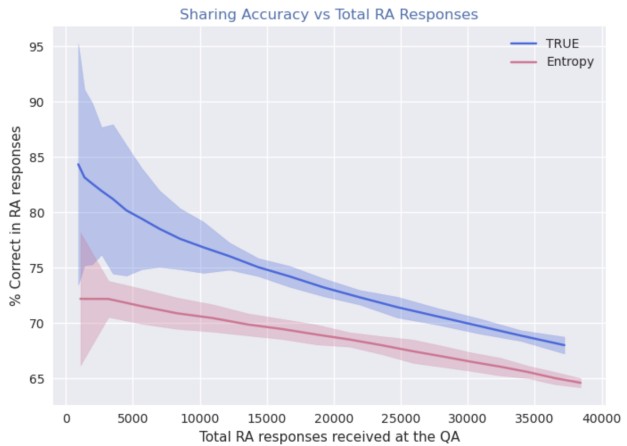

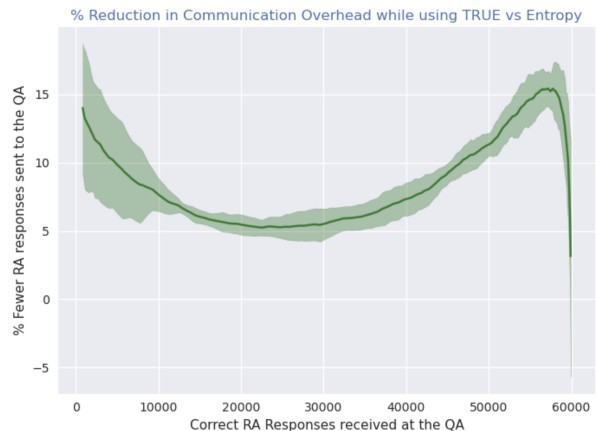

(a) TRUE vs Entropy: This plot displays the percentage of correct responses in the responses sent by RAs to the QA using TRUE and Entropy as confidence scores (high is better). TRUE outperforms Entropy consistently.

(b) A PEEPLL system using TRUE needs 5-20% fewer total RA responses (vs. Entropy) for the QA to receive the same number of correct responses. This contributes to a reduction in system communication overhead.

Figure 10: These plots correspond to evaluation on MiniImageNet.

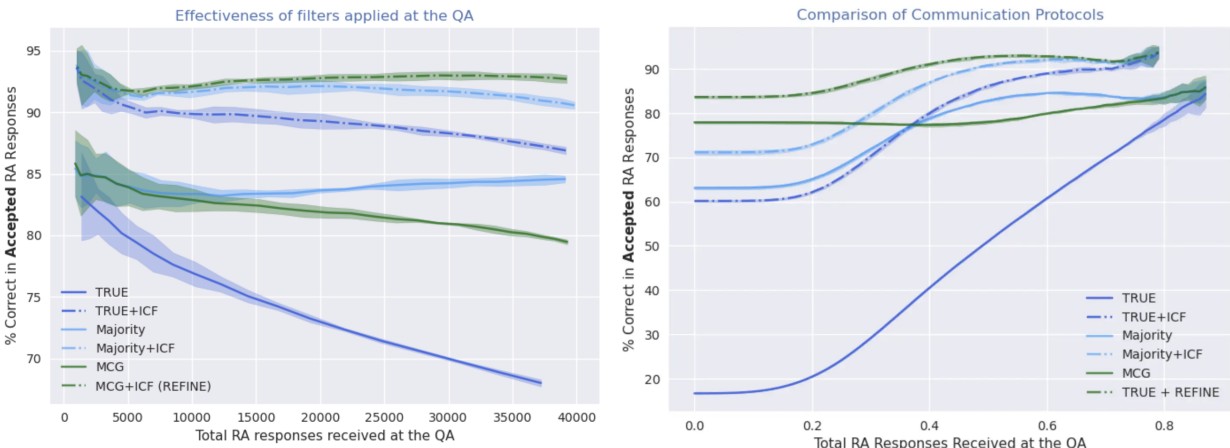

Figure 11: MiniImageNet: Each filter selectively accepts a subset of the responses received at the QA from the responses dispatched by the RAs utilizing TRUE. The figure depicts the ratio of correct responses within the selected subset. In the left figure, we cut off at a threshold of a 1:2 ratio of Queries asked to Responses received for clarity.

# E Computation Complexity & Scalability

## E.1 Symbols

$\mathbf{M}$: Memory size, $\mathbf{K}$: Total number of classes in the system, $\mathbf{K(i, t)}$: Number of classes in agent $i$ at time $t$, $\mathbf{N(q, out)}$: Number of agents queried for query q, $\mathbf{N(q, back)}$: Number of agents responding back for query q, $\mathbf{N}$: Number of agents, $\mathbf{M(i, k, t)}$: Number of samples in memory of class $k$ at time $t$ for agent $i$, $\mathbf{d}$: Dimension of latent vector, $\mathbf{C}$: Cost of communication, $\mathbf{F_N}$: Forward pass of the entire network (Encoder + Decoder), $\mathbf{F_E}$: Forward pass of the Encoder, $\mathbf{B_N}$: Backward pass of the entire network (Encoder + Decoder).

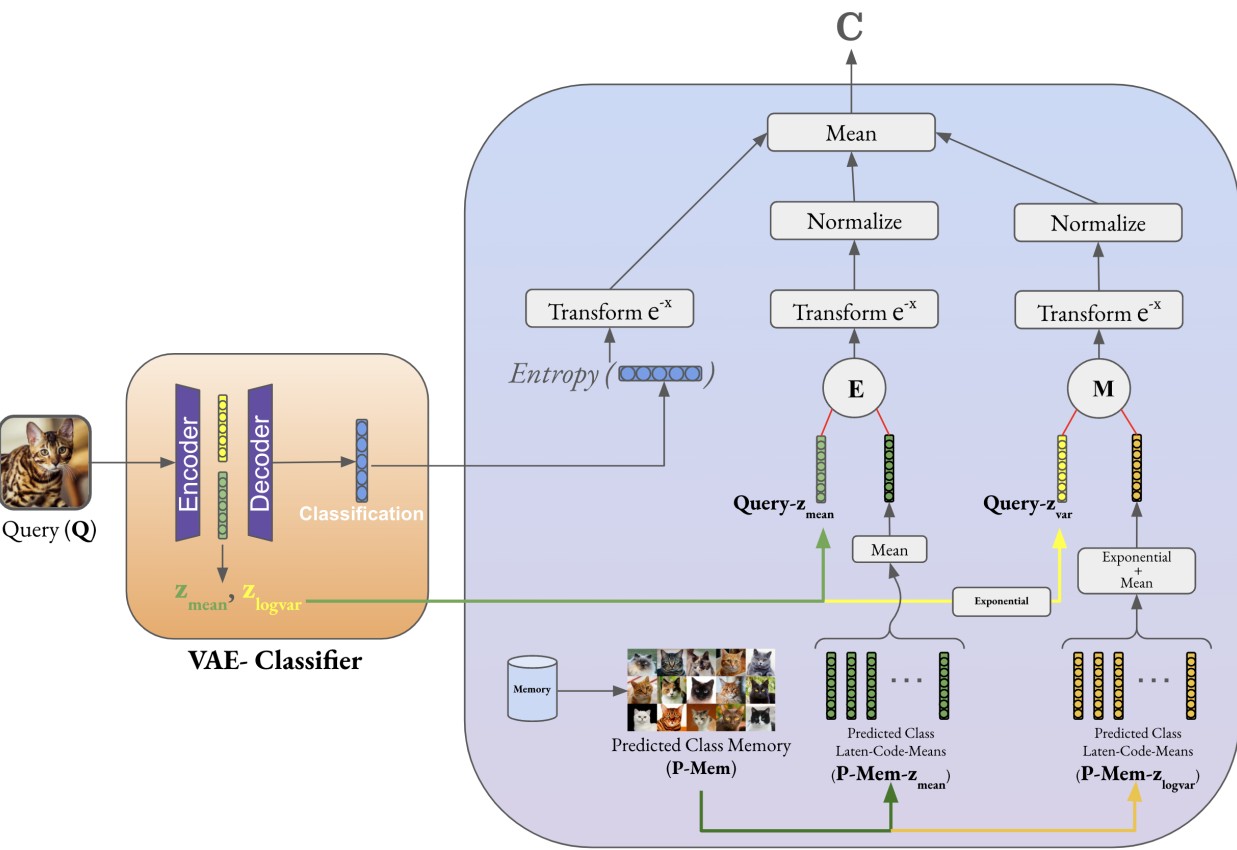

Figure 12: TRUE: Confidence Evaluation mechanism for an agent. The circle with E denotes the operation that returns the Euclidean distance between two vectors, and M the Manhattan Distance. 'Transform $e^{-x}$' takes in a value and maps it using the $e^{-x}$ function.

Note $N(q, back) \leq N(q, out) \leq N$. The number of agents chosen for querying is an important research question. This selection should not be based on their previous interactions. Instead, it could be determined by factors such as proximity or other metadata.

### E.2 Computational Complexity Analysis

### E.3 Query Agent Complexity

**Confidence Calculation**

- *Get the prediction class k*: $O(\mathrm{F}_N)$

- *Find memory samples belonging to k*: $O(1)$

- *Evaluate latent codes for those memory samples*: $O(\mathrm{F}_E \times M(i, k, t))$

- *Calculate Dispersion and Semantic Distance*: $O(d \times M(i, k, t))$

- *Get the mean of Dispersion and Semantic Distance*: $O(1)$

- *Transform Mean Dispersion and Semantic Distance*: $O(1)$

- *Get Entropy of Prediction*: $O(d)$

- *Transform Entropy*: $O(1)$

- *Average three scores*: $O(1)$

- **Final Complexity:** $O(F_N + (F_E + d) \times M(i,k,t))$

## Seeking Assistance

- *Check confidence against threshold*: $O(1)$

- *Send query and confidence to other agent*: $O(N(q,out) \times C)$

- **Final Complexity:** $O(N(q, out) \times C)$

## Selective Filtering

- *Put agents into groups by their predicted class*: $O(N(q,back))$

- *Calculate each group's confidence (MCG)*: $O(1)$ (keep track while sorting)

- *Pick group with highest confidence (MCG)*: $O(1)$ (keep track while sorting)

- *Pick group with highest members (Majority)*: $O(1)$ (keep track while sorting)

- *Reject conflicting responses (ICF)*: $O(N(q,back) \times K)$* or $O(N(q,back) \times N(q,back))$**.

- **Final Complexity:** $\text{Minimum}(O(N(q,back) \times K), O(N(q,back) \times N(q,back))$ when ICF is used. Else, $O(N)$.

  *For a query, make a dictionary with keys as classes and values as the predictions of the agents who know that class. Maintain this dictionary as you receive the answers $O(N(q,back) \times K)$. Now, go through each response, and if the predicted class of that response has any value in the dictionary that is not the same as the key (predicted class), disregard that response $O(N(q, back))$. Hence, the complexity would be $O(N(q,back) \times K)$.

  **Go through each response and then check each agent to see if they know the response's predicted class, and if they do, compare their predictions with the prediction of the response. If they do not match, reject the response. The complexity is $O(N(q,back) \times \sum_i k(i,t)) = O(N(q,back) \times N(q,back) \times K)$, but if known classes in agents are indexed by the class id in a dictionary, we have $O(N(q,back) \times \sum_i 1)$ and the algorithm becomes $O(N(q,back) \times N(q,back))$.

## Learning Update

- *Backward pass*: $O(B_N)$

## Final QA Complexity:

- Minimum of the following

  $O(F_N + (F_E + d) \times M(i,k,t) + N(q,out) \times C + N(q,back) \times K + B_N)$

  $O(F_N + (F_E + d) \times M(i,k,t) + N(q,out) \times C + N(q,back) \times N(q,back) + B_N)$

### E.4 Response Agent Complexity

- *Confidence Evaluation:* $O(F_N + (F_E + d) \times M(i,k,t))$

- *Sending Response Back:* $O(C)$

- **Final RA complexity:** $O(F_N + (F_E + d) \times M(i,k,t) + C)$

### E.5 Scalability Analysis

**Total Complexity for Agents to exist in PEEPLL**    For $Q$ queries: For an agent i, of the Q queries, Q1 (i) is the number of times agent i serves as QA, and Q2 (i) is the number of times it serves as RA. Note that Q1(i) + Q2(i) ≤ Q. Therefore, the total complexity is

$$Q1(i) \cdot (F_N + (F_E + d) \times M(i,k,t) + N(q,out) \times C + N(q,back) \times K + B_N) + Q2(i) \cdot (F_N + (F_E + d) \times M(i,k,t) + C)$$

Generalizing the above by considering the three terms that scale, Q, N, and K, we have a complexity of **O(Q×N(., out)×K)**, where N(., out) is the number of agents that it reaches out to for any query.

## F    Design Decisions

1. **Entropy in TRUE:** Confidence scores derived from Semantic and Dispersion distances alone outperform Entropy by 21.2% on CIFAR100 and 6.5% on MiniImageNet. This is comparable to TRUE (Semantic + Dispersion + Entropy) vs Entropy. Entropy provides an additional stream of information that may be beneficial for some applications.

2. **Semantic and Dispersion Distances:** Dhuliawala et al. (2024); Prasad et al. (2020); Alemi et al. (2016); Keel et al. (2023) show that the position of data in the latent space holds significance to the class it belongs to. To further support this, we plot the latent space and demonstrate the formation of clusters according to classes, as also recently shown by Prasad et al. (2020); see Figure 13. In TRUE, we measure the divergence in the mean and variance that define a datapoint's position in the latent space. Intuitively, TRUE is representative of how close a data point would be to the cluster predicted by the model in the latent space. Their inclusion is supported by ablation studies (Section 4.1, TRUE Results, para 1).

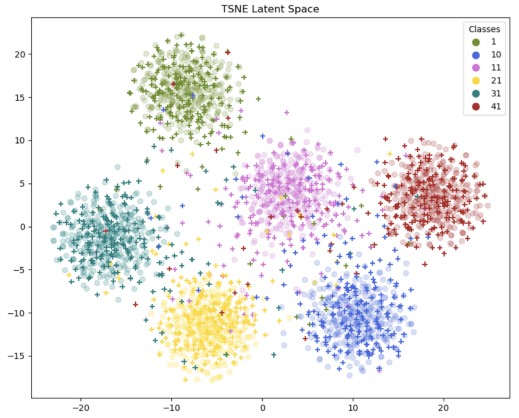
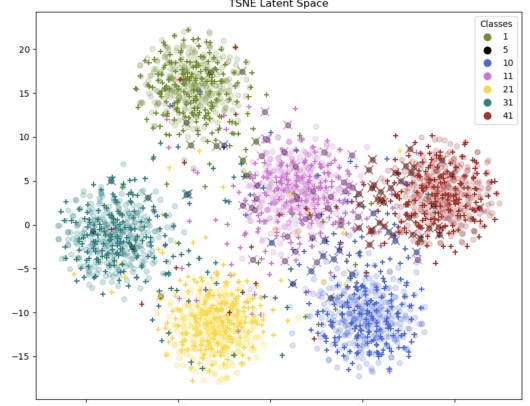

(a) Lighter dots represent the training data, and '+' denotes the test data. The colors of the dots and '+' represent its true labels. The agent has an average TRUE confidence of 0.75 for predicting the '+' data.

(b) 'x' represents data of class 5, which the model is not trained on. The color of 'x' indicates the model's prediction. The agent has an average TRUE confidence of 0.34 for predicting the 'x' data.

Figure 13: An agent's t-SNE visualization of its learned latent distribution of classes it is trained on in CIFAR100. The agent is trained on classes displayed in Figure 1. Observe that the 'x' data is generally toward the edges of the clusters, and the 'x' latent vectors are not clustered themselves.

