# OpenReview forum: "Distributed Multi-Agent Lifelong Learning"
_TMLR — Accepted by TMLR_

### Review · Reviewer_sb6p · 2024-07-26

**Summary Of Contributions:**

The paper studies lifelong learning in a multi-agent environment. The authors propose a distributed lifelong learning framework in which agents can continually learn online with other agents via query-reponse mechanism. They show that the communication among agents scales well in terms of computational efficiency. In experiments, the authors demonstrate the resilience of communication and the efficiency of memory usage.

**Audience:**

Yes

**Claims And Evidence:**

No

**Requested Changes:**

See them in Weaknesses.

**Strengths And Weaknesses:**

Strengths

- The authors propose a multi-agent lifelong learning framework called Peer, Parallel Lifelong Learning, which allows agents to improve learning performance without environmental supervision.

- The authors establish some benchmark components of the proposed framework and demonstrate that it can outperform environmental supervision.

Weaknesses

- The proposed framework intuitively generalizes lifelong learning to multiple agents. Although the framework appears to be a new concept, specific technical developments are not entirely clear.

- The computational complexity and the scalability of the proposed method are not characterized.

- The proposed method lacks a theoretical foundation, leaving performance questionable.

---

> ### Author Response · Authors · 2024-08-04
>
> We sincerely thank the reviewer for taking out the time to give meaningful feedback. Your suggestions were especially helpful in strengthening the theoretical basis our paper.
>
> $\textbf{W1:}$ Thank you for acknowledging the conceptual novelty of our framework. We clarified our technical contributions in the "Contributions" section at the end of Section 1.
>
> Our study introduces three major technical advancements, each through a dedicated algorithm to the components of our framework: (1) TRUE (confidence score): This superior uncertainty quantification method outperforms traditional entropy-based measures (see paper's Figure 2). (2) ICF: A highly effective new filtering algorithm (see Table 1). (3) Dynamic Memory-Update: A memory management algorithm focused on system resilience (last paragraph of Section 4.3).
> These algorithms promote stability in learning and reduce communication overhead in PEEPLL.
>
> TRUE is also a general contribution to the ML community for uncertainty quantification, while ICF for the Multi-Agent ML community.
>
> $\textbf{W2:}$ This is a great addition to the paper. We find the following computation complexity for an agent in PEEPLL, in terms of the scalable symbols - Q total queries, N(., out) the number of RAs a QA contacts for a query, and K total classes -
>
> $\textbf{O}$$\textbf{(Q}$ $\times$ $\textbf{N(., out)}$ $\times$ $\textbf{K)}$.
>
> We provide the full details of the derivation of the computational complexity in the [attached PDF](https://drive.google.com/file/d/1FpqYTsPRKbpGWxuyLYXCJ89kSLu9RYY0/view?usp=sharing).
>
> In our updated manuscript, we have included a comment at the end of Section 4.1 (second to last paragraph) and a complete derivation in Appendix's Section E, Computation Complexity & Scalability. This strengthens our analysis of the proposed solution to PEEPLL. Thank you for the suggestion!
>
> $\textbf{W3:}$ Thank you for your feedback - we now add citations and a figure (attached below) to strengthen the theoretical basis of TRUE.
>
> We would like to clarify that our study is experimental, and we carefully evaluate our proposed methods through empirical evidence. We support each method with experimental results, ablation studies, and thorough analysis, drawing parallels to existing literature.
>
> (1) Lifelong Learning Performance: In addition to the empirical evidence, Section 4.3 (Paragraph 4) explains how learning under PEEPLL parallels Domain-IL, where Lifelong Learning is easier [1]. This advantage supports the superior performance of a PEEPLL agent over a single agent with ER.
>
> (2) TRUE Performance: We added - "[2, 3, 4, 5] show that the ***position*** of data in the latent space holds significance to the class it belongs to. To further support this, we plot (see new [attached figure](https://drive.google.com/file/d/136JsI4c8kwT8RzUN-XiREMA0PTthXmjy/view?usp=drive_link)) the latent space of an agent and demonstrate the formation of clusters according to classes, as also shown by the very recently proposed [2]. In TRUE, we simply measure the divergence in the *properties* that define a datapoint's position in the latent space (properties: mean and variance). Intuitively, TRUE is representative of how close a data point would be to the cluster the model predicts it belongs to in the latent space." - to the Appendix Section F Design Decisions, point 2 of our updated manuscript.
>
> $\textbf{References:}$
>
> [1] van de Ven, G.M., Tuytelaars, T. & Tolias, A.S. Three types of incremental learning. Nat Mach Intell 4, 1185–1197 (2022).
>
> [2] Dhuliawala, S. Z., Sachan, M., & Allen, C. (2024). Variational classification: A probabilistic generalization of the softmax classifier. Transactions on Machine Learning Research.
>
> [3] Alemi, A., Fischer, I., Dillon, J., & Murphy, K. (2017). Deep variational information bottleneck Int. In Conf. on Learning Representations (Vol. 3).
>
> [4] Prasad, Vignesh et al. “Variational Clustering: Leveraging Variational Autoencoders for Image Clustering.” 2020 International Joint Conference on Neural Networks (IJCNN) (2020): 1-10.
>
> [5] Keel, Benjamin et al. “Variational Autoencoders for Feature Exploration and Malignancy Prediction of Lung Lesions.” British Machine Vision Conference (2023).

---

### Review · Reviewer_rvfH · 2024-07-29

**Summary Of Contributions:**

The paper proposes a new system to perform multi-agent lifelong learning. Roughly speaking, each agent makes predictions on inputs, but also receives inputs from other agents to refine their predictions. The system consists of three components: 1) agents evaluate confidence of predictions, 2) agents filter responses from other agents, 3) agents use a memory bank to enable continual learning. The paper evaluates each of these components in isolation and together and demonstrate that the components and the system overall outperform baselines on image classification tasks.

**Audience:**

Yes

**Broader Impact Concerns:**

No broader impact concerns.

**Claims And Evidence:**

No

**Requested Changes:**

**Critical for acceptance**

Compare with additional baselines in continual learning and multiagent learning
Please fix citations (change /cite to /citep when appropriate)

**Would strengthen**

Include a design appendix detailing each design decision made in the paper

**Strengths And Weaknesses:**

**Strengths**

The proposed system uses many novel components and each of these components is carefully evaluated. It's clear the authors put a lot of thought into the design of the system, and this comes through in the presentation of the paper.

The paper is well-written overall, and all components of the algorithm are well motivated. The experimental setup is carefully described with relevant details. Plots are nice with error bars and good color choices.

**Weaknessess**

In my view, the main weakness of the paper is the lack of effective baselines. The main baselines used by the authors are a single agent, and different versions of the proposed framework. First of all, the single agent underperforms the proposed method due to lack of regularization in the single agent. I would encourage the authors to regularize the single agent until it outperforms the proposed method to have a fair comparison.

More importantly, there are no other continual learning/lifelong learning/multiagent methods against which the method is compared. I understand that the authors may be designing an algorithm for a setting that is not considered in prior literature; nevertheless, it is important to include results from additional methods even if they are not applicable to the setting. For instance, how would a single agent with experience replay perform? How about multiple agents with a shared experience replay pool? It's ok if these methods outperform the proposed PEEPLL system, but it's useful to get a sense of how well the system performs in comparison to stronger baselines.

Also, the proposed system is fairly complex. The authors test many components of their system in isolation, but given the large number of design choices, it may be infeasible to empirically evaluate each one. Unfortunately, without empirical isolation, these design choices can appear as arbitrary heuristics rather than the careful choices they are. I encourage the authors to include a comprehensive design appendix that details the motivation and pros/cons of each of the design choices made by the authors (for example, why is the form of dispersion disparity in terms of exp(log var) instead of log var directly, why is entropy chosen as a component of TRUE, etc.).

The authors state that they use a VAE in the confidence evaluation strategy. However, the proposed method doesn't use a reconstruction loss, so it's by definition not an autoencoder. It might be more accurately described as a classifier with a particularly constrained latent representation. It's not at all clear to me how the authors would have considered this to be a VAE.

The ICF description in section 3.2 could be expanded upon a little bit more.

There are some formatting issues in the paper as well: in certain places, parenthetical citations should be used instead of in-line citations.

---

> ### Author Response · Authors · 2024-08-08
>
> We sincerely appreciate your careful evaluation. Your suggestions have been particularly helpful in adding meaningful baselines and better positioning our work.
>
> **Critical for Acceptance:**
>
> **Q1:** “Compare with additional baselines in continual learning and multiagent learning”
>
> **Done!** We previously had one single-agent baseline: Experience Replay (ER), which is state-of-the-art for single agents LL ([1]), and two multi-agent baselines: PEEPLL+Entropy and PEEPLL performance with Static Memory-Update (Section 4.3, last para).
>
> We now include **additional 9 baselines**; see [attached table](https://drive.google.com/file/d/1I9QmLm-tdQ1I1B-Gm543DuoFRM6REua3/view?usp=sharing) (rows 1-8, 10), also see Table 2 in the updated manuscript:
> 1. Upper-bound baselines for LL (both Single and Multi-Agent): IID online & offline.
> 2. Single-agent baselines: EWC, ER (state-of-the-art), GDumb (very strong baseline).
> 3. Lower-bound baselines for Single-Agent: Online & Offline Naive.
> 4. Lower-bound baseline for Multi-Agent: PEEPLL+Refine - Naive.
> 5. Strong baseline for Multi-Agent LL: GDumb (detailed below how GDumb is also relevant for multi-agent LL).
>
> Notes:
> 1. We incorporated your suggestion to evaluate a shared experience replay pool among agents using GDumb ([2]). We implemented it with a 100% accurate communication protocol, where the QA retrains its network from scratch using shared memory.
> 2. IID samples independently and identically from all of the training data.
>
> **Q2:** “/cite to /citep when appropriate” - **Done!**
>
> **Other Comments:**
>
> **Q1:** “regularize the single agent”
>
> **A1:** We appreciate your suggestion and experimented with adding L2 regularization, using weights ranging from 1e-3 to 1e-6. Unfortunately, this did not lead to any improvement in performance.
>
> The superior performance of PEEPLL agents compared to the single agent with ER, despite PEEPLL agents learning from potentially incorrect responses, may seem counterintuitive. In LL, agents sequentially learn tasks with new classes, which could lead to forgetting old tasks, as seen with the Single-Agent with ER. Under PEEPLL, the agent briefly learns about future tasks through incorrect responses from RAs (Figure 5a, 8a), similar to Domain-Incremental Learning, where agents sequentially learn tasks with new data but no new classes. This is also evidenced by the QA’s increasing confidence at the beginning of each subsequent task (Figure 6a). Domain-IL-like learning makes it easier for agents to learn and retain knowledge ([1]), giving our PEEPLL agent an advantage. We analyze this in detail in Section 4.3, para 4.
>
> **Q2:** “Design Appendix“
>
> **A2:**  We are happy to do so. We have included a Section F. Design Appendix in the updated manuscript.
>
> "1. Entropy in TRUE: Confidence scores derived from ...
>
> 2. Semantic and Dispersion Distances: [3, 4, 5] show that the ***position*** of data in the latent space holds significance to the class it belongs to. To further support this, we [plot](https://drive.google.com/file/d/136JsI4c8kwT8RzUN-XiREMA0PTthXmjy/view?usp=sharing) ..."
>
> Further, we also discussed the choice of distances in the TRUE metric in Section 3.1, para starting with “The choice of Euclidean distance…”
>
> Note: Since the latent code is sampled from Mean and Variance, we measure divergence in the property of Variance. We believe using log_var would work too (only on a different scale).
>
> **Q3:** “not an autoencoder”
>
> **A3:** Fixed; we now refer to it as a *Variational Classifier*. It appears our architecture was very recently proposed by [3] and named *Variational Classifier*. We mention this in Section 3.1. Our uncertainty quantification remains novel as an application of a Variational Classifier.
>
> **References:**
>
> [1] Gido M. van de Ven, Tinne Tuytelaars, and Andreas S. Tolias. Three types of incremental learning. Nature Machine Intelligence
>
> [2] Prabhu, Ameya et al. “GDumb: A Simple Approach that Questions Our Progress in Continual Learning.” European Conference on Computer Vision (2020).
>
> [3] Dhuliawala, S. Z., Sachan, M., & Allen, C. (2024). Variational classification: A probabilistic generalization of the softmax classifier. Transactions on Machine Learning Research.
>
> [4] Alemi, A., Fischer, I., Dillon, J., & Murphy, K. (2017). Deep variational information bottleneck Int. In Conf. on Learning Representations.
>
> [5] Keel, Benjamin et al. “Variational Autoencoders for Feature Exploration and Malignancy Prediction of Lung Lesions.” British Machine Vision Conference (2023).

---

### Review · Reviewer_HNSr · 2024-09-25

**Summary Of Contributions:**

The authors propose a framework called Peer Parallel Lifelong Learning (PEEPLL), where agents continuously learn online by seeking assistance from other confident agents, rather than relying on human annotators. PEEPLL consists of three key components: 1) confidence evaluation, 2) selective filtering, and 3) lifelong learning. Notably, the confidence evaluation is based on the newly introduced TRUE score, proposed by the authors. The TRUE score leverages a variational autoencoder to enhance the confidence assessment of other agents. The proposed framework demonstrates significant performance improvements on CIFAR-100 and MiniImageNet, and the ablation study clearly highlights the effectiveness of each component.

**Audience:**

Yes

**Broader Impact Concerns:**

No significant ethical concerns are identified.

**Claims And Evidence:**

Yes

**Requested Changes:**

**Requested Changes**
Since the primary goal of lifelong learning is to enable models, such as LLMs, to acquire new information that arises over time, illustrating the application of this broad framework could significantly strengthen the paper. Additionally, while the TRUE score shows substantial improvements over evaluation techniques using the entropy, it is highly dependent on the model architecture. Therefore, the authors should discuss these limitations in the paper.

**Strengths And Weaknesses:**

**[Strengths]**
The paper is well-written and easy to follow. The proposed lifelong learning framework efficiently utilizes other agents without relying on human annotations. The TRUE score effectively improves lifelong learning performance compared to simple entropy-based confidence evaluation.

**[Weaknesses]**
While the authors claim that the principles of PEEPLL are applicable across various domains in multi-agent lifelong learning, it is not clear how this would work in areas such as large language models (LLMs). Since all experiments are conducted on supervised image classification tasks (CIFAR-100 and MiniImageNet), the scalability of the framework to other domains is not sufficiently demonstrated. The applications of lifelong learning framework need to be clearly explained.

A key limitation of the TRUE score is the reliance on a specific architecture (i.e., variational autoencoder). Given that many advanced perception tasks use discriminative architectures, adding a probabilistic regularization term might harm performance. Moreover, the TRUE score cannot be applied directly to pre-trained foundation models since it requires training with a probabilistic regularization term, which is typically not used during the training of widely used foundation models.

---

> ### Author Response · Authors · 2024-09-26
>
> We sincerely appreciate your careful evaluation. Your insights (1) helped us expand on how PEEPLL’s principles can be applied to other domains like language modeling, robotics, and medical AI, and (2) allowed us to discuss TRUE’s scope.
>
> **Q1:** Applicability of PEEPLL’s principles to other domains.
>
> **A1:** We have added the following text to the beginning of the PEEPLL section 1.1 in the updated manuscript:
>
> “PEEPLL’s mechanism and framework are designed to navigate the dynamic, uncertain environments typical of multi-agent LL settings; it focuses on judicious peer communication without overwhelming the system with excessive communication. Multi-agent cooperation across domains of fields helps reduce reliance on annotated data (environmental supervision). Regardless of the domain, in any decentralized multi-agent LL system, agents must be self-aware—able to autonomously identify novel tasks, actively seek help, responsibly assist others, and carefully evaluate received help.
>
> While this study focuses on supervised image classification, Lifelong Learning is also increasingly being applied in domains like language modeling [1, 2], robotics [3, 4], and medical AI [5, 6], where agents develop sub-domain-expertise in their respective environments through continual learning. The principles underlying PEEPLL are applicable across various domains. In language modeling, for instance, PEEPLL can allow LLM agents to communicate with peers when handling uncertain prompts, allowing them to continuously learn to generate accurate and contextually relevant responses. The received peer responses can also be used for future fine-tuning. In high-stakes medical AI, PEEPLL can allow agents to seek assistance when uncertain and take only confident and well-validated actions. In robotics, PEEPLL can facilitate real-time collaboration, enabling robots to responsibly exchange sub-domain-specific expertise and adapt to new tasks (either online or offline, depending on the application). The **Framework** section below provides domain-agnostic guidelines for developing dedicated algorithms to the components of PEEPLL.”
>
>
> **Q2:** Limitation of TRUE.
>
> **A2:** This is important to include; thank you. We included the following text at the end of Section 4.1 in the updated manuscript:
>
> "While TRUE has demonstrated clear advantages over entropy-based methods for uncertainty quantification, we note that its reliance on a variational classifier limits its direct application to pre-trained models and purely discriminative architectures. This presents interesting opportunities to advance and expand TRUE’s applicability to such models by modifying the architecture and fine-tuning."
>
> **References:**
>
> [1] Gururangan, Suchin et al. “Don’t Stop Pretraining: Adapt Language Models to Domains and Tasks.” ArXiv abs/2004.10964 (2020): n. Pag.
>
> [2] Qin, Yujia et al. “Recyclable Tuning for Continual Pre-training.” Annual Meeting of the Association for Computational Linguistics (2023).
>
> [3] Lesort, Timothée et al. “Continual Learning for Robotics.” ArXiv abs/1907.00182 (2019): n. Pag.
>
> [4] Logacjov, Aleksej et al. “Learning Then, Learning Now, and Every Second in Between: Lifelong Learning With a Simulated Humanoid Robot.” Frontiers in Neurorobotics 15 (2021): n. Pag.
>
> [5] Lee, Cecilia S, and Aaron Y Lee. “Clinical applications of continual learning machine learning.” The Lancet. Digital health vol. 2,6 (2020): e279-e281. doi:10.1016/S2589-7500(20)30102-3
>
> [6] Vokinger, Kerstin N et al. “Continual learning in medical devices: FDA's action plan and beyond.” The Lancet. Digital health vol. 3,6 (2021): e337-e338. doi:10.1016/S2589-7500(21)00076-5

---

### Author Response · Authors · 2024-09-27

**Overall Response**

We thank everyone involved for taking the time to provide meaningful insights that improved the overall quality of our submission.

We appreciate the positive views expressed by reviewers that our work is **carefully evaluated** to highlight the effectiveness of each component (rvfH, HNSr), develops **many novel components** (rvfH), including the **conceptual novelty of the framework** (sb6p, rvfH, sb6p), and finally that our paper is **well-written** (HNSr, rvfH).

In our responses to the reviewers below, we addressed their precise concerns directly and provided an updated submission pdf. In our updated submission, we have highlighted in *blue* portions of the text, which have been modified to allow for quick validation by the reviewers and AE. To summarize the main changes we have made to the submission:

* we have expanded on how PEEPLL can be applied to other domains
* we have added additional baselines in continual learning and multiagent continual learning
* we have included a Design Decisions appendix to support the choices made while developing TRUE
* we have analyzed and added the computation complexity for an agent in PEEPL

---

### Decision · Action_Editor_7GDS · 2024-12-23

**Recommendation:** Accept with minor revision

**Comment:**

Overall, I think this paper focuses on the experimental results of a new framework and provides enough evidence to support the effectiveness of the design. The authors provide the source code for the experiments in this paper.

In the minor revision, I suggest the authors to fix the typos in this paper and improve the readability and comprehensiveness of the source code.

---

All reviewers acknowledge that the design of this framework is novel, and the paper is well-written.

Reviewer sb6p questioned about 'specific technical developments' and 'computational complexity and the scalability' which are answered by the authors in the paper. Reviewer sb6p also cares about the theoretical foundation of this paper which is still a weakness of this paper.

Reviewer rvfH questioned about 'effective baselines' and 'evaluation of each component in this complex system', which are answered by the authors in the paper.

Reviewer HNSr questioned about the application of this framework in other domains such as large language models. This is still a weakness of this paper (based on the current experiments) although the authors have updated the manuscript to explain the applicability.

**Audience:**

People who want to design a distributed multi-agent lifelong learning will be interested in this paper (for the framework, the experiment setup, and the results).

**Claims And Evidence:**

This paper proposed a novel framework, Peer Parallel Lifelong Learning (PEEPLL), for solving distributed Multi-Agent Lifelong Learning. The three components in PEEPLL are (1) Confidence-Evaluation (supported by Figure 2), (2) Selective Filtering (supported by Table 1 and Figure 3), (3) Lifelong Learning in PEEPLL (supported by Table 2). The reduction of system communication overhead is supported by Figure 4. All current evidence are from empirical results based on the experiments designed by the authors, while this paper still lacks of theoretical support.

---

> ### Author Response · Authors · 2025-01-10
> **Submitted Camera-Ready Version**
>
> Hello Dr. Wang,
>
> We have completed the minor revisions as requested:
>
> **Text:** We made minor changes to improve clarity and readability, including adding a brief introductory background on Lifelong Learning in the Related Works section. None of our edits altered the meaning of any sentences.
>
> **Code:** We modularized the codebase and added comments throughout the files to help readability. In particular, we have provided detailed comments on the implementations of our TRUE, introduced filters, and the Dynamic Memory-Update.
>
> Please let us know if there are any additional adjustments you would like us to make.
>
> Finally, we sincerely thank you for your valuable feedback and support throughout this process, and we also sincerely thank the reviewers for their feedback. Collectively, your insights truly helped improve the quality of the paper.